# Microstructure and Impact Toughness of Laser-Arc Hybrid Welded Joint of Medium-Thick TC4 Titanium Alloy

Peng Luo [1,2], Wanxi Feng [2], Gang Zu [3], Linyin Luo [2] and Jun Xiao [1,*]

1 College of Materials Science and Technology, Nanjing University of Aeronautics and Astronautics (NUAA), Nanjing 210016, China; luopeng_1983@163.com
2 Avic General Huanan Aircraft Industry Co., Ltd., Zhuhai 519040, China
3 Tang Shan Iron and Steel Group Co., Ltd., Tangshan 063000, China
* Correspondence: j.xiao@nuaa.edu.cn

**Abstract:** This study delves into the impact toughness of medium-thick (12 mm thick) titanium alloy joints crafted through a multi-layer, multi-pass welding technique that blends laser-arc (MIG) hybrid welding technology. Microstructural scrutiny, employing optical microscopy, SEM and TEM, unveils a consistent composition across weld passes, with prevailing α/α′ phases interspersed with some β phase, resulting in basket-weave structures primarily dominated by acicular α′ martensite. However, upper regions exhibit Widmanstatten microstructures, potentially undermining joint toughness. Hardness testing indicates higher values in cosmetic layers (~420 HV) compared to backing layers and bending tests manifest superior toughness in lower joint regions, attributed to smaller grain sizes induced by repetitive welding thermal cycles. Impact toughness assessment unveils diminished values in the weld metal (WM) compared to the heat-affected zone (HAZ) and base material (BM), amounting to 91.3% of the base metal's absorption energy. This decrement is ascribed to heightened porosity in upper regions and variations in grain size and phase composition due to multi-layer, multi-pass welding. Microstructural analysis proximal to failure sites suggests one mechanism wherein crack propagation is impeded by the β phase at acute crack angles. In essence, this study not only underscores the practicality of laser-MIG hybrid welding for medium-thick TC4 alloy plates but also underscores the reliability of joint mechanical properties.

**Keywords:** titanium welding; laser-MIG hybrid welding; microstructural analysis; impact toughness; medium-thick welded joint



## 1. Introduction

Titanium alloys have gained significant attention in various industries due to their exceptional mechanical properties, including a high strength-to-weight ratio, excellent corrosion resistance, and biocompatibility [1–3]. These properties make titanium alloys ideal for critical applications such as aerospace components, medical implants, and sporting equipment. Especially in the actual industrial generation and manufacturing, such as shipbuilding, aviation, and other fields, the medium-thick cross-section TC4 titanium alloy is being used more and more widely [4,5]. For this reason, in the manufacturing process of some large complex structures, TC4 alloys with medium-thick sections need to be welded for practical use. However, welding titanium alloys presents challenges due to their high reactivity with atmospheric gases and susceptibility to welding metal (WM) and heat-affected zone (HAZ) embrittlement [1,6,7]. Traditional metal inert gas (MIG) arc welding techniques often result in limited weld penetration depth, narrow weld width, and the formation of weld defects, such as porosity and cracking, particularly in medium-thick or thick sections of titanium alloys [8,9].

To overcome these challenges and enhance the mechanical properties of welded joints, researchers have explored various welding techniques, including gas tungsten arc

welding (GTAW), plasma arc welding (PAW), electron beam welding (EBW), and laser beam welding (LBW) [1,10–13]. Among these techniques, laser beam welding has emerged as a promising method for joining titanium alloys due to its high energy density, minimal heat input, and precise control over the welding process [14,15]. Furthermore, several researchers have explored the integration of artificial intelligence algorithms and alternative strategies to enhance the monitoring and adjustment of the welding process [16,17]; yet, challenges persist in the welding of titanium alloys, specifically concerning these process issues. To mitigate these constraints, innovative hybrid welding techniques, merging laser welding with supplementary welding methods, have been introduced. Notably, the laser-MIG hybrid welding technique has emerged as a significant advancement, offering enhanced penetration depth, increased weld width, and improved modulation of weld microstructure [18]. In addition, the use of arc bridging ability can achieve a synergistic effect of "1 + 1 > 2". Laser-arc hybrid welding only needs a few layers of thick plate, which significantly improves the efficiency compared with traditional MIG welding. By combining the advantages of both laser and MIG welding processes, laser-MIG hybrid welding offers a potential solution to overcome the challenges associated with traditional laser welding techniques and produce high-quality welded joints in titanium alloys [7,18].

In recent years, there has been a growing interest in utilizing laser-MIG hybrid welding for titanium alloys. Several studies have explored the microstructural characteristics, mechanical properties, and optimization of process parameters in laser-MIG hybrid welded titanium alloy joints [1,7]. For instance, Su et al. [19] investigated the impact of heat input (E) on the microstructure and tensile strength of 15 mm thick TC4 alloy plate joints fabricated through laser-MIG hybrid welding. They observed that increasing heat input prolonged the residence time of $\alpha'$ martensite in the high-temperature phase transformation region, leading to thicker $\alpha'$ phases and increased β thickness, along with the presence of dislocations in $\alpha'$. Moreover, higher heat input resulted in reduced tensile strength. Brandizzi et al. [20] examined the laser-MIG hybrid welding process for 3 mm thick TC4 alloy and identified the optimal welding parameters by analyzing the cross-section morphology of the weld. Compared to laser welding, laser-MIG hybrid welding achieved well-formed welds with lower heat input, mitigating the risk of weld collapse observed in laser welding. Chen et al. [21] explored the influence of defocusing amount on porosity defects in laser-MIG welding of titanium alloy. Their welding simulation revealed that welding pores primarily resulted from keyhole collapse. Increasing the defocusing amount reduced laser energy density, shallowed the keyhole depth, and weakened the eddy current of the molten pool metal, suppressing keyhole-type pore defects. Fan et al. [22] conducted single-pass welding tests on 5 mm thick TC4 alloy plates using laser-TIG hybrid welding. Their findings demonstrated that increasing welding current improved the tensile strength of the welded joint. Yang et al. [23] performed butt welding tests on 6 mm thick titanium alloy plates by adjusting process parameters in laser-arc hybrid welding. They found that laser-arc hybrid welding allowed for higher welding speed and lower heat input compared to traditional arc welding. Additionally, the laser-induced concentration of arc energy improved microhardness by refining grain structures in the WM and HAZ. Li et al. [14] compared laser-MIG hybrid welding and laser welding joints of 4 mm Ti-Al-Zr-Fe alloy, revealing that the former method produced joints with higher ductility. Optimal parameter settings ensured defect-free welds with no surface oxidation, pores, cracks, or insufficient penetration. Laser-MIG hybrid welding also demonstrated significant advantages in joining dissimilar titanium alloy-gold alloys. Gao et al. [24] utilized laser-arc hybrid welding to connect dissimilar Ti6Al4V titanium alloy with AISI316 stainless steel, investigating the effects of welding parameters on weld bead shape, microstructure, mechanical properties, and fracture behavior.

However, scholars mostly focus on the study of thin titanium alloys. For large thick plates, some scholars adopt narrow-gap laser-arc hybrid welding and beam shaping to improve the weld formation and performance [25,26], but there are still few studies on the laser-MIG hybrid welding parameters and their effects on the joint performance of TC4

medium-thick cross-section alloy, so further studies are needed [9]. In this study, the feasibility and mechanical performance of laser-MIG hybrid welding for joining medium-thick titanium alloy plates is investigated. Specifically, this study focuses on the microstructural characteristics, hardness distribution, and impact toughness of the welded joints fabricated using a multi-layer, multi-pass laser-MIG hybrid welding technique. By comprehensively characterizing the microstructure and mechanical properties of the welded joints, this research aims to provide insights into the optimization of laser-MIG hybrid welding parameters and the enhancement of joint performance in titanium alloy applications.

## 2. Experimental Details

### 2.1. Material

The base material (BM) used is TC4, also known as Ti-6Al-4V, which is one of the $\alpha + \beta$ titanium alloys. This material, with a thickness of 12 mm, was fabricated through the hot rolling process by Baoti Co. Ltd., Baoji, China. The dimensions of the test specimens are $150 \times 100 \times 12$ mm. TC4 wire with a diameter of 1.2 mm is used for welding, with its composition detailed in Table 1. The mechanical properties of both the BM and the wire are presented in Table 2.

**Table 1.** Element of TC4 alloy.

| Element | Al | V | Fe | C | N | H | O | Ti |
|---|---|---|---|---|---|---|---|---|
| TC4 alloy | 6.2 | 4.05 | 0.13 | 0.016 | 0.007 | 0.0008 | 0.131 | Bal. |
| TC4 welding wire | 5.96 | 4.01 | 0.10 | 0.012 | 0.007 | 0.0001 | 0.126 | Bal. |

**Table 2.** Physical properties of TC4 alloy at 20 °C.

| | Yield Stress $R_{p0.2}$ (MPa) | Ultimate Stress $R_m$ (MPa) | Elongation A (%) |
|---|---|---|---|
| TC4 alloy | 922 | 986 | 13.8 |
| TC4 welding wire | 895 | 993 | 15.5 |

### 2.2. Laser-Arc Hybrid Welding System

The primary equipment utilized during the welding process is depicted in Figure 1. The laser-arc hybrid welding system, as shown in Figure 1a, consists of a FANUC robot (FANUC, Oshino-mura, Japan), an IPG Photonics YLS-10000 laser (IPG, New York, NY, USA) with a 10 kW maximum power, 0.6 mm spot diameter, 300 mm focal length, and a Fronius TPS4000 arc welder (Fronius International GmbH, Pettenbach, Austria). As illustrated in Figure 1b, the laser-MIG hybrid welding process for medium-thickness plates adopts a leading laser and trailing arc configuration, with the laser beam angled at approximately 85° horizontally and 60° between the laser and the arc. The distance between the laser beam and the arc is approximately 2.5 mm. A shroud containing 99.999% pure argon gas is employed during welding to prevent adverse reactions between titanium alloy and atmospheric elements at high temperatures, ensuring weld quality. Additionally, high-purity argon serves as the backing gas for the weld.

The dimensions of the laser-MIG hybrid welded titanium alloy specimens are illustrated in Figure 2a, featuring a zero gap, 5 mm blunt edge, and double-sided bevel angle of 30°. Figure 2b depicts the welding sequence and pass distribution, comprising the 1st#, 2nd#, and 3rd# passes corresponding to the back weld, packing weld, and cosmetic weld layers, respectively. The welding process parameters for the weld joint, determined through preliminary testing and adjustments, are detailed in Table 3.

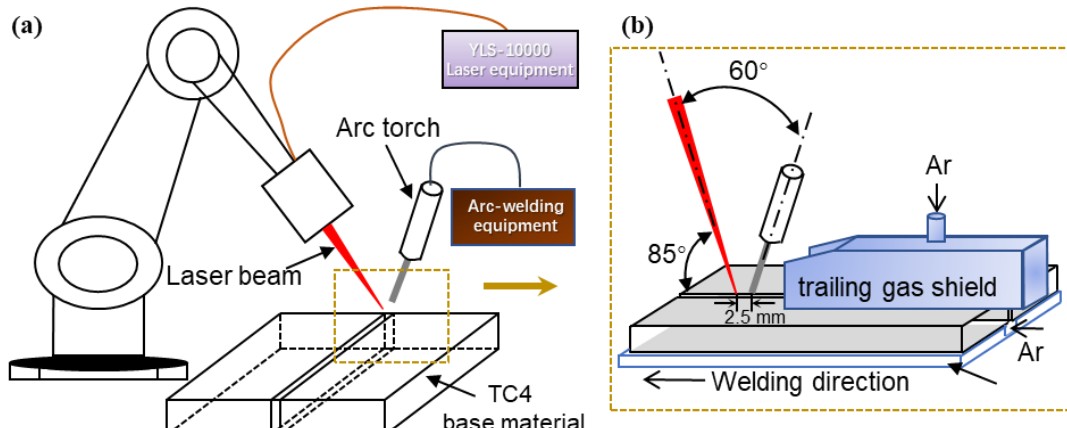

**Figure 1.** (**a**) Schematic diagram of the laser-arc hybrid welding process; (**b**) Schematic diagram of TC4 plate butt welding.

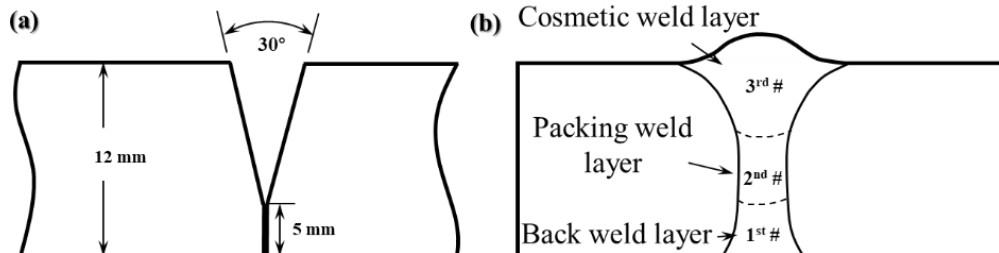

**Figure 2.** (**a**) Schematic diagram size of laser-MIG hybrid TC4 joint; (**b**) schematic diagram of welding sequence and distribution of welding passes.

**Table 3.** Parameters of TC4 alloy laser-MIG hybrid welding process.

| Number | Welding Pass | Laser Power (kW) | Welding Speed (mm/s) | Wire Feed Rate (m/min) |
|--------|--------------|------------------|----------------------|------------------------|
| 1st # | Back weld layer | 4500 | 10 | 10 |
| 2nd # | Packing weld layer | 4000 | 10 | 9 |
| 3rd # | Cosmetic weld layer | 6000 | 16 | 15 |

*2.3. Microstructure, Hardness, Bending and Impact Properties Tests*

The overview diagram of the sampling position of weld properties is shown in Figure 3. Each pair of plates should be tested in the middle of the weld as far as possible.

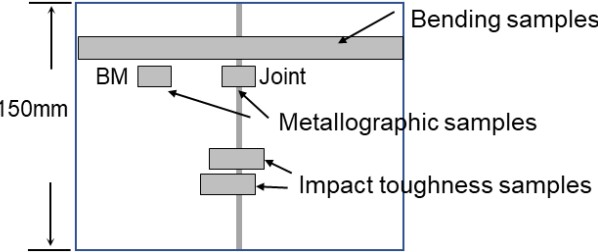

**Figure 3.** Schematic diagram of sampling position.

2.3.1. Microstructure Characterization

To analyze the characteristics and properties of the laser-arc hybrid welded titanium alloy joint, the microstructure of the joint and BM samples requires characterization and analysis. Initially, metallographic samples are obtained by wire cutting perpendicular to the

weld seam, followed by grinding with sandpaper, polishing, and etching in Keller's reagent (HF:HNO$_3$:H$_2$O = 1:3:16) for approximately 25~30 s. Subsequently, the microstructure of the joint is observed and analyzed using a Zeiss optical microscope (OM, ZEISS temi, Shanghai, China), a scanning electron microscope (SEM, ZEISS Gemini 300, Oberkochen, Germany), and a transmission electron microscope (TEM, FEI Tecnai F20, Thermo Fisher Scientific, Waltham, MA, USA). Among them, the TEM samples were prepared using an EM Precision Cutter (Model EM-PC300, Shanghai, China) to slice ultrathin sections. These sections were then thinned to electron transparency using an ion milling machine (Model IM4000, Shanghai, China). The final thickness was monitored under a light microscope to ensure optimal electron transmittance for TEM analysis.

### 2.3.2. XRD Test

The XRD analysis was conducted using a Panalytical Empyrean diffractometer equipped with a Cu Kα radiation source. The scanning speed was set to 1°/min, covering a 2θ range from 30° to 80°. This setup ensures comprehensive coverage and detailed data collection for accurate phase identification and lattice parameter calculations.

### 2.3.3. Hardness Test

The schematic diagram of hardness testing positions is shown in Figure 4, utilizing an HVS-20 Vickers microhardness tester, with tests conducted in accordance with GB/T 4340.1 [27]. During testing, the ambient temperature is maintained at approximately 20 °C, with a load of 1 kg applied for 15 s and a step size of approximately 0.5 mm for each indentation.

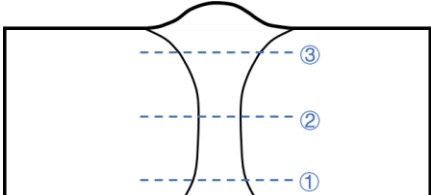

**Figure 4.** Schematic representation of the position of hardness tests.

### 2.3.4. Bending Test

Bend testing, a common method for assessing mechanical properties, is conducted to evaluate the plasticity and bonding strength between the weld and the BM, with tests carried out in accordance with GB/T 2653 standards [28]. Rectangular bend specimens measuring 200 × 20 × 10 mm are cut from the test plates, and the specimens are fixed during testing. The three-point bending apparatus, with a bending core diameter of approximately 40 mm and capable of bending angles up to 180°, is used to apply a continuous force at the midpoint between the supports until the specimen bends to the specified angle or exhibits visible cracking. The same parameter should be tested at least 3 times, and the corresponding data should be processed.

### 2.3.5. Impact Toughness Test

The impact toughness testing equipment used is the PTM220 impact tester, with tests performed at least 3 times once a weld layer in accordance with GB/T 2650 standards [29]. Figure 5 illustrates the sampling scheme for impact test specimens of the titanium alloy welded joint, with several specimens taken along a plane perpendicular to the weld direction from both the HAZ and WM. Figure 5b provides a schematic diagram of the specimen dimensions, with the impact fracture surfaces observed using an SEM (ZEISS Gemini 300, Oberkochen, Germany).

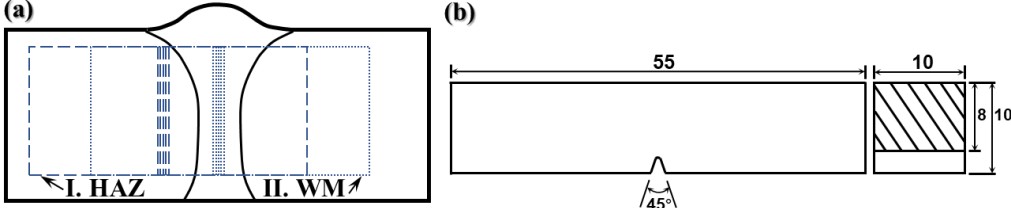

**Figure 5.** (**a**) Diagram of positions of impact toughness test specimens; (**b**) size of specimens (mm).

## 3. Results and Discussion

### 3.1. Microstructure Analysis

The microstructure of the BM is illustrated in Figure 6, and it is observed longitudinally from the rolling direction perpendicular to the plate, where the lighter areas represent equiaxed primary $\alpha$ phases, and the darker areas depict lamellar $\alpha + \beta$ phases distributed along the edges of the $\alpha$ phase, with an average grain size of 5 $\mu$m.

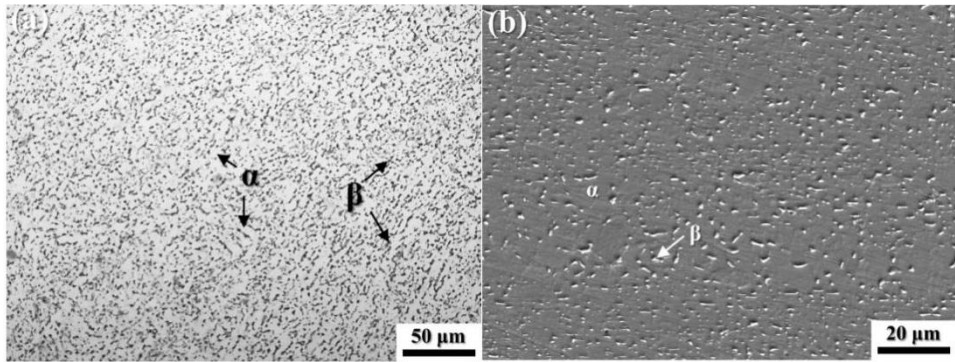

**Figure 6.** Microstructure of TC4 base metal by: (**a**) OM; (**b**) SEM.

Figure 7a displays the metallographic cross-section of the titanium alloy laser-MIG hybrid welded joint, where the upper and lower surfaces are ground to approximately "10 mm" thickness. The WM and HAZ exhibit an overall V-shaped nail-like characteristic, with the filler layer in the middle occupying a smaller area distribution proportion. Moreover, the grain size of the weld bead in the 3rd# layer is noticeably larger than that at the bottom (1st# layer). Within the WM, there are a few equiaxed grains, and numerous columnar grains exist on both sides of the weld, with these columnar grains nucleating at the interface between the BM and the weld and growing toward the center of the weld. During the solidification process of the molten pool, the temperature at the center of the weld is higher than at the sides, causing the grains to grow along the temperature gradient and in the direction of columnar grain growth. As the columnar grains grow, the solid-liquid interface moves toward the center of the weld, increasing the undercooling. Meanwhile, the temperature at the center of the weld decreases over time, inhibiting the growth of existing grains and promoting the formation of new grains, ultimately leading to the formation of uniformly sized equiaxed grains [27]. In Figure 7b, the macroscopic morphology of the root and cosmetic layer weld beads on both the upper and lower surfaces under these parameter conditions is presented. The weld bead surfaces appear silver-white without any oxidation phenomenon, and there is minimal spatter on the weld bead surface, indicating overall good formability.

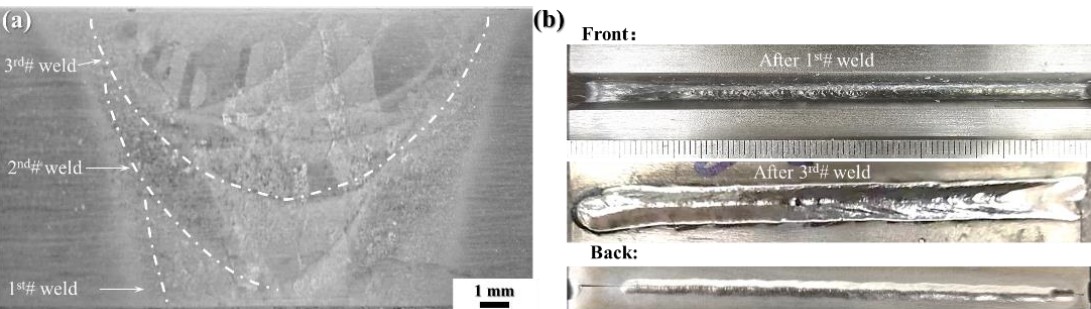

**Figure 7.** (**a**) The microstructure distribution of the welded joint; (**b**) Macroscopic morphology of the weld.

Figure 8 depicts the microstructure and distribution of the weld bead in the 3rd# cosmetic layer of the joint at low magnification (OM) and high magnification (SEM) under a microscope. In Figure 8a, the grain size within the joint ranges from 200 to 600 μm, with numerous $\alpha'$ martensite needle-like structures distributed within the $\beta$ primary grain boundaries. Figure 8b illustrates the approximately vertical distribution of $\alpha'$ martensite needle-like structures, forming a basket-weave microstructure. This is attributed to the rapid cooling during the welding process, leading to the transformation of the $\beta$ phase into $\alpha'$. Due to the high nucleation rate of $\alpha'$ in different orientations, the size of $\alpha'$ clusters decreases, and the $\alpha'$ needles widen and shorten within the original columnar grains. The $\alpha'$ martensite nucleates and grows within the original columnar grains, intertwining with each other, forming needle-like martensite with a certain orientation relationship, and terminating upon encountering grain boundaries.

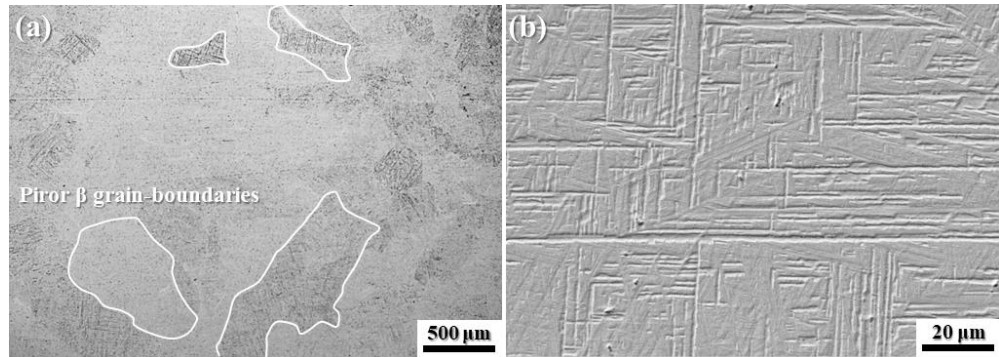

**Figure 8.** Microstructure of WM of the laser-MIG hybrid welded TC4 joint: (**a**) OM; (**b**) SEM.

Figure 9 illustrates the microstructural morphology of the WM and HAZ within the corresponding regions of the three weld passes in the joint. Due to slower cooling rates at the center of the weld bead, the surrounding columnar grains exert a certain hindrance to temperature reduction. Consequently, after heating and temperature elevation, there is minimal change in temperature once it decreases to a certain extent, leading to rapid grain coarsening in the WM and the appearance of some larger equiaxed grain structures. Comparing Figure 9a,c,e, it is evident that the grain size in the 3rd# layer weld bead is significantly larger than that in the previous two passes. The width of the columnar grains is approximately 400 μm in the 3rd# layer weld bead, whereas the grain size in the 1st# layer weld bead is only about 220 μm, with the majority being equiaxed grains. This is primarily attributed to the different welding thermal cycles experienced by the weld beads in different layers.

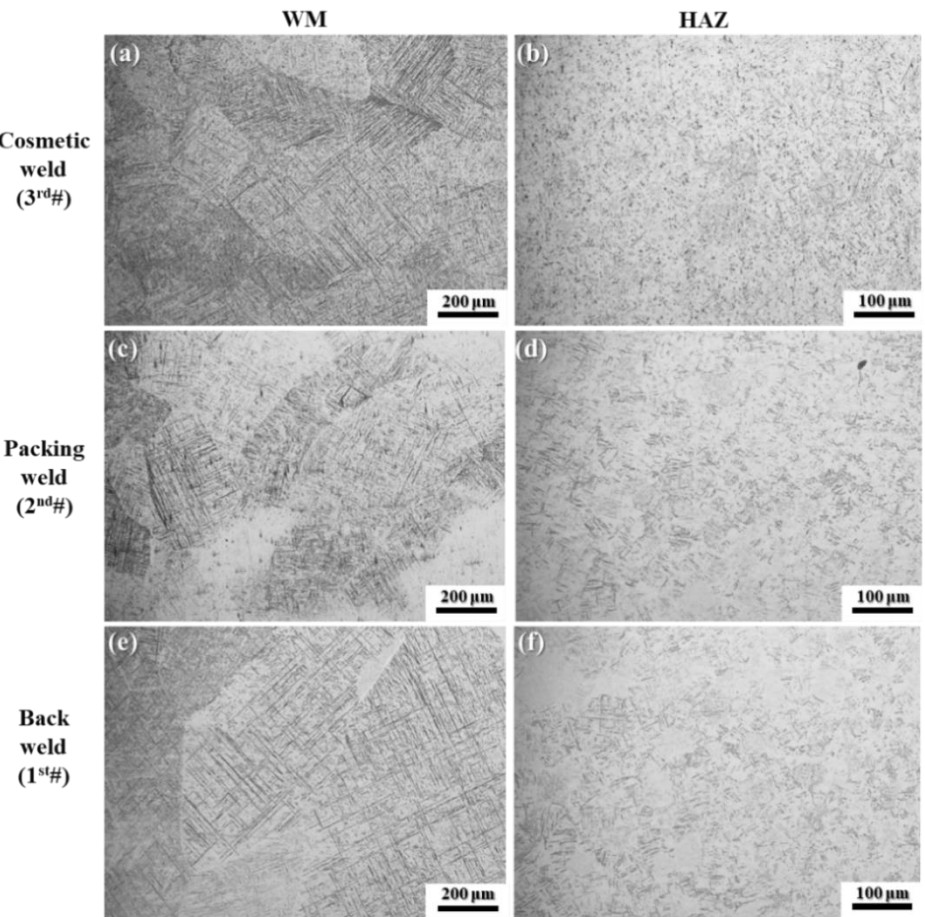

**Figure 9.** Microstructure of the different zones of laser-MIG hybrid welded TC4 joint. (**a**) WM in 3rd# weld; (**b**) HAZ in 3rd# weld; (**c**) WM in 2nd# weld; (**d**) HAZ in 2nd# weld; (**e**) WM in 1st# weld; (**f**) HAZ in 1st# weld.

The bottom weld bead undergoes a second heating-cooling cycle while being welded over the upper layers, with the peak temperature relatively lower compared to the first cycle [28]. However, it is still possible to reach the temperature for β→α′ transformation. Due to the secondary thermal cycle, the cooling rate accelerates compared to the previous cycle, preventing the grains from growing. This effectively anneals the bottom weld bead, to some extent enhancing the uniformity of its microstructure distribution. Basket-weave structures are observed in all three layers, with coarse Widmanstatten microstructures formed near the β grain boundaries in the WM of the 2nd# and 3rd# layers. These structures result from the rapid cooling after the transformation from the primary β phase, above the β → α′ transformation point, forming either α′ lamellae or α′ + β lamellae within the grains.

The microstructural morphology of the HAZ at the three locations exhibits minimal differences in size. The HAZ is influenced by the heat from the WM, reaching the temperature for α phase to β phase transformation. Rapid grain growth occurs in the β phase, with smaller primary β grains near the WM due to slower cooling rates and longer dwell times at high temperatures. After cooling, the microstructure transforms into α′ martensite, forming an interlocking basket-weave structure. In areas closer to the BM, where the cooling rates are faster, the dwell time in the β phase is shorter, resulting in finer α′ grains. In the transition zone between the HAZ and the BM, the dwell time in the β phase is even shorter, leading to the formation of secondary α′, with some untransformed α phases fused together to form blocky α phases [29]. Upon completion of cooling, a mixture of secondary α′ and blocky α phases forms the microstructure.

### 3.2. XRD Analysis

Figure 10 presents the micro-area XRD spectra of different weld passes within the same weld bead. There are no significant differences in grain orientation among the different layers, with the majority consisting of $\alpha/\alpha'$ phases. The β phase only appears at the peak (110), which is close to the position of the $\alpha'$ peak (0002). Compared to the two layers below the joint, the diffraction intensity of the hexagonal crystal system grains is relatively higher. Particularly, in the 3rd# cosmetic layer weld bead, the diffraction intensity of the (10–11) crystal orientation is significantly higher, indicating a distinct orientation of $\alpha/\alpha'$ phases in the cosmetic layer. This is attributed to the differences in microstructural distribution and size generated during the solidification process of the molten pool, as well as the fewer thermal cycles experienced by the upper weld bead during the multi-pass welding process compared to the lower layers.

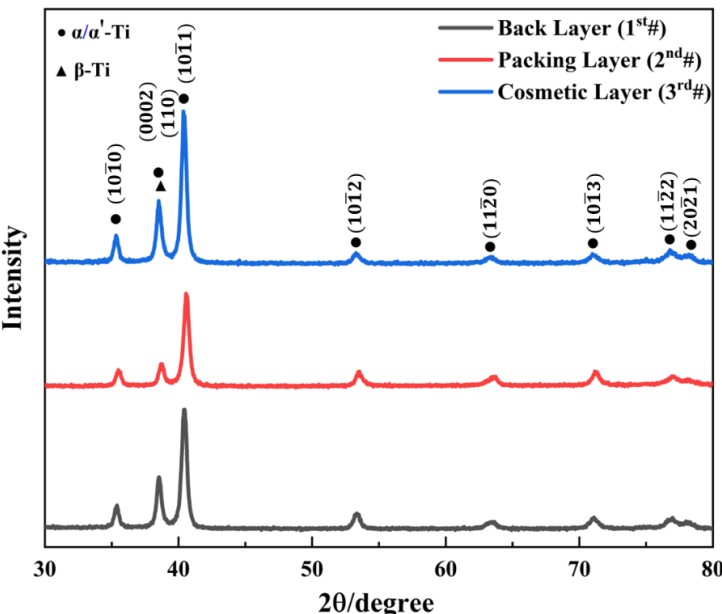

**Figure 10.** XRD results of different zones of the welded joint.

### 3.3. Mechanical Properties of the Joint

Hardness Analysis

The hardness curve, as depicted in Figure 11, exhibits a convex profile, indicating that the hardness in the vicinity of the weld bead's center is significantly higher than that in the BM. As is depicted, Figure 11a–c shows the horizontal hardness test results of the three layers of the weld, respectively. Near the HAZ, the hardness reaches a relative peak, gradually decreasing toward the HAZ and the BM transition. Due to the close elemental composition between the weld bead and the BM and the gradual decrease in fusion degree, the energy for nucleation on semi-melted grains in the fusion zone is minimal. Therefore, during the solidification of the weld pool, nuclei are first formed near the fusion zone and grow toward the center of the weld pool in the form of columnar grains. As the columnar grains continue to grow and migrate, they push lower-melting-point solutes or impurities toward the center of the weld pool. Consequently, due to the segregation effect, a large amount of relatively large $\alpha'$ phases are generated within the weld bead, resulting in slightly lower hardness values in the middle of the weld bead compared to the HAZ. Vertically, the hardness value is highest in the cosmetic weld (3rd#), reaching approximately 440 HV. The increased hardness in WM of titanium alloy welding is primarily due to the microstructure formed by rapid cooling during the welding process. In the WM, rapid cooling can lead to the formation of fine, hardened $\alpha'$martensitic phases. This fine martensitic structure is much harder than the original structure of the titanium alloy, thus

increasing the overall hardness of the welded area [30]. This is also related to the observed distribution of Widmanstatten structures [31,32]. The overall variation in hardness values of the 2nd# and 1st# weld passes is less pronounced compared to the upper layer (3rd#), with hardness values in the weld bead zone approximately 420 HV.

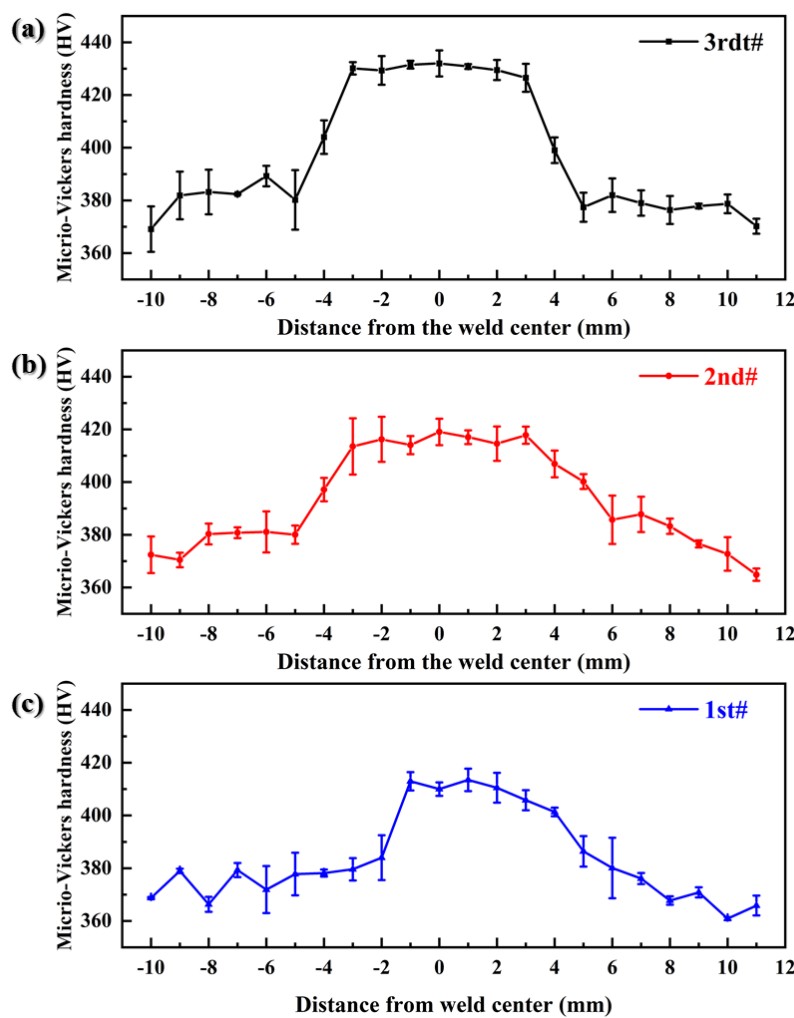

**Figure 11.** Hardness distribution of laser-MIG hybrid welded TC4 alloy joint: (**a**) 3rd# layer, (**b**) 2nd# layer, (**c**) 1st# layer.

### 3.4. Bending Test

Table 4 provides the test results for both face bend and root bend tests. The data in groups 1–3 correspond to the micro-zone positions sampled from the upper (3rd#), middle (2nd#), and lower (1st#) parts of the joint, respectively. It can be observed that the test forces in the root bend tests are generally slightly higher than those in the positive bend tests. Particularly in the third group of specimens, the test force in the root bend test reaches 31.41 kN, which is slightly higher than the highest value of 30.10 kN in the positive bend test. The standard deviations of the test forces for positive bending and back bending are 0.08 and 0.69, respectively. This indicates that the joint's load-bearing capacity under root bending loading is slightly higher than that under face bending loading, indicating better toughness performance. The bending results reveal that the bending performance of the 3rd# weld bead is superior to that of the 1st# and 2nd# weld beads, which may be attributed to the microstructure size in the WM region.

**Table 4.** Results of the bending test.

| Number | Mode | Bending Angle (°) | Bending Force (kN) | Average Force (kN) | Standard Deviation |
|---|---|---|---|---|---|
| 1-1 | | 9.4 | 30.05 | | |
| 2-1 | Face bending | 14.8 | 29.90 | 30.02 | 0.08 |
| 3-1 | | 15.7 | 30.10 | | |
| 1-2 | | 9.6 | 31.41 | | |
| 2-2 | Root bending | 13.2 | 31.49 | 30.96 | 0.69 |
| 3-2 | | 25.1 | 29.98 | | |

This disparity in bending performance can be correlated to the microstructural characteristics observed within the WM. Specifically, the superior performance of the 3rd# weld bead suggests a finer and more homogeneous microstructure, which is known to enhance the toughness and load-bearing capacity of the material under stress. Furthermore, the observed standard deviations imply a more consistent mechanical behavior in the face bend tests compared to the root bend tests, potentially indicating a more uniform distribution of residual stresses and microstructural features across the tested zones. Future studies could further elucidate these relationships by correlating microstructural features with mechanical properties across different sections of the weld joint, as has been suggested in the literature [33,34]. This comprehensive approach can offer deeper insights into the influence of welding parameters on joint performance.

*3.5. Room-Temperature Impact Toughness*

3.5.1. Impact Toughness Analysis

The results of the room-temperature impact performance in different micro zones of the joint are shown in Table 5. The impact absorption energy in the WM is minimal, only 30.6 J, while in the HAZ, it is 33.3 J, reaching a level close to that of the BM (33.5 J). The WM remains the most brittle part of the joint in terms of toughness, whereas the HAZ exhibits better impact toughness compared to the WM, likely due to smaller dimensions and a smoother transition in grain size. These findings suggest that HAZ demonstrates enhanced toughness properties, likely due to the thermal cycling experienced during the welding process, which may refine the grain structure or alter the phase composition, thus improving the impact resistance. The BM, with an impact absorption energy of 33.5 J, shows the highest toughness, indicating that the original material properties are largely preserved outside of the direct influence of welding heat. Additionally, the variance in impact absorption energy, represented by the standard deviations, indicates a level of inconsistency within the zones, with the WM area showing a higher variability (standard deviation of 2.18) compared to the HAZ (standard deviation of 1.90). This variation could be attributed to factors such as welding speed, temperature gradients, and material flow during the welding process, which can introduce heterogeneities in the microstructure, particularly in the WM. The difference between the WM and HAZ impact toughness also highlights the right combination can minimize the adverse effects of welding on the microstructure and enhance the overall performance of the welded joint [35,36].

**Table 5.** Results of impact toughness test in different locations. (Unit of $A_{kv}$: J).

| Number | WM | HAZ | BM |
|---|---|---|---|
| 1 | 33.1 | 34.0 | 34.0 |
| 2 | 31.0 | 30.7 | 33.0 |
| 3 | 27.8 | 35.2 | 33.5 |
| Average (J) | 30.6 | 33.3 | 33.5 |
| Standard Deviation | 2.18 | 1.90 | 0.41 |

Figure 12 illustrates the macroscopic morphology of the impact fracture surfaces corresponding to the results in Table 5. It can be observed that the fracture surfaces of all three

regions consist of a fibrous zone near the notch, a bright and radially patterned radiative zone, and shear lips extending along the 45° direction, displaying typical characteristics of ductile fracture. The fracture surface in the BM appears relatively flat and smooth, without obvious defects. In the HAZ, some minor pores are present in the radiative zone, while the shear lips exhibit some larger pores. In the WM, both the radiative zone and the shear lips feature pores of notably differing sizes (approximately 30–50 μm), exhibiting a clustered distribution, particularly as observed in Figure 12(a2,a3). This phenomenon primarily stems from the collapse and closure of keyhole pores during the welding process, culminating in process-induced porosity [9,37]. Such pores are likely to diminish the impact absorption energy of the WM, consequently reducing its toughness and resulting in a divergence from the anticipated optimal weld toughness. The specimens in group number 1 exhibit better impact toughness in all regions, with the fracture surface being the flattest, smoothest, and showing clear ductile fracture characteristics, with the least number of pores and defects.

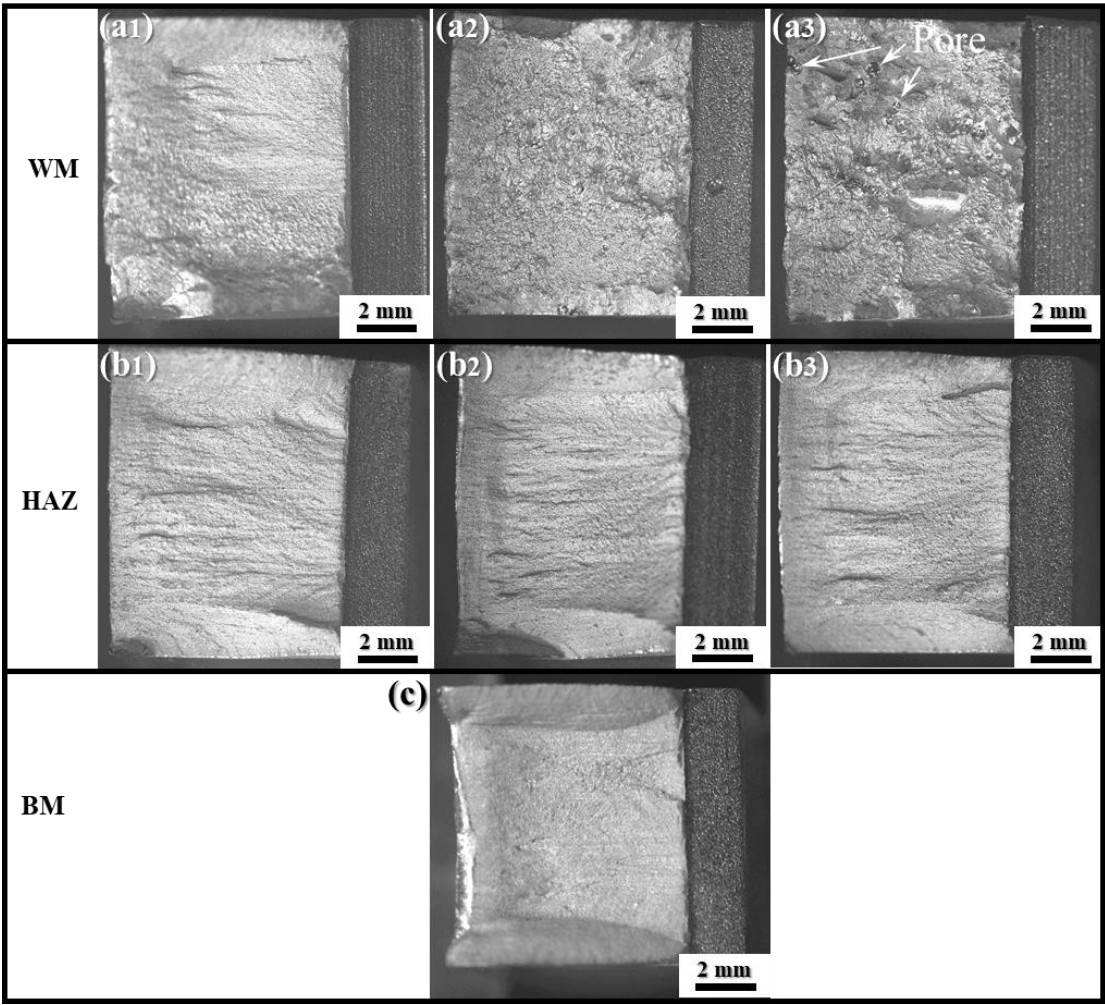

**Figure 12.** Macroscopic fracture morphology of laser-MIG hybrid welded TC4 joint: (**a1**–**a3**) sampled WM as the center; (**b1**–**b3**) sampled HAZ as the center; (**c**) sampled BM as the center.

The microstructure characterization of the impact fracture surface in the WM, which exhibits better impact toughness, from the first group of samples is depicted in Figure 13(a1,a2). The fracture surface in the fibrous zone shows numerous and unevenly sized dimples, indicating a ductile-brittle mixed fracture. These dimples, formed by the accumulation of microvoids or pores under the action of external force, exhibit ductile fracture characteristics. The radiative zone displays deep micropores clustered in a river-like pattern, along with some larger micropores that represent tearing dimples. The parabolic direction of the

tearing dimples indicates the crack source, accelerating local ductile fracture. The central part of the fibrous zone is mainly composed of unevenly sized equiaxed dimples, indicating good plasticity, with a small portion showing river-like patterns. The interweaving distribution of numerous dimples and a few river-like patterns suggests a ductile fracture, indicating relatively good toughness. Additionally, there is a large shallow pit and a large micropore, which have minimal impact on the overall impact performance. The fibrous zone exhibits similar-sized and evenly distributed dimples.

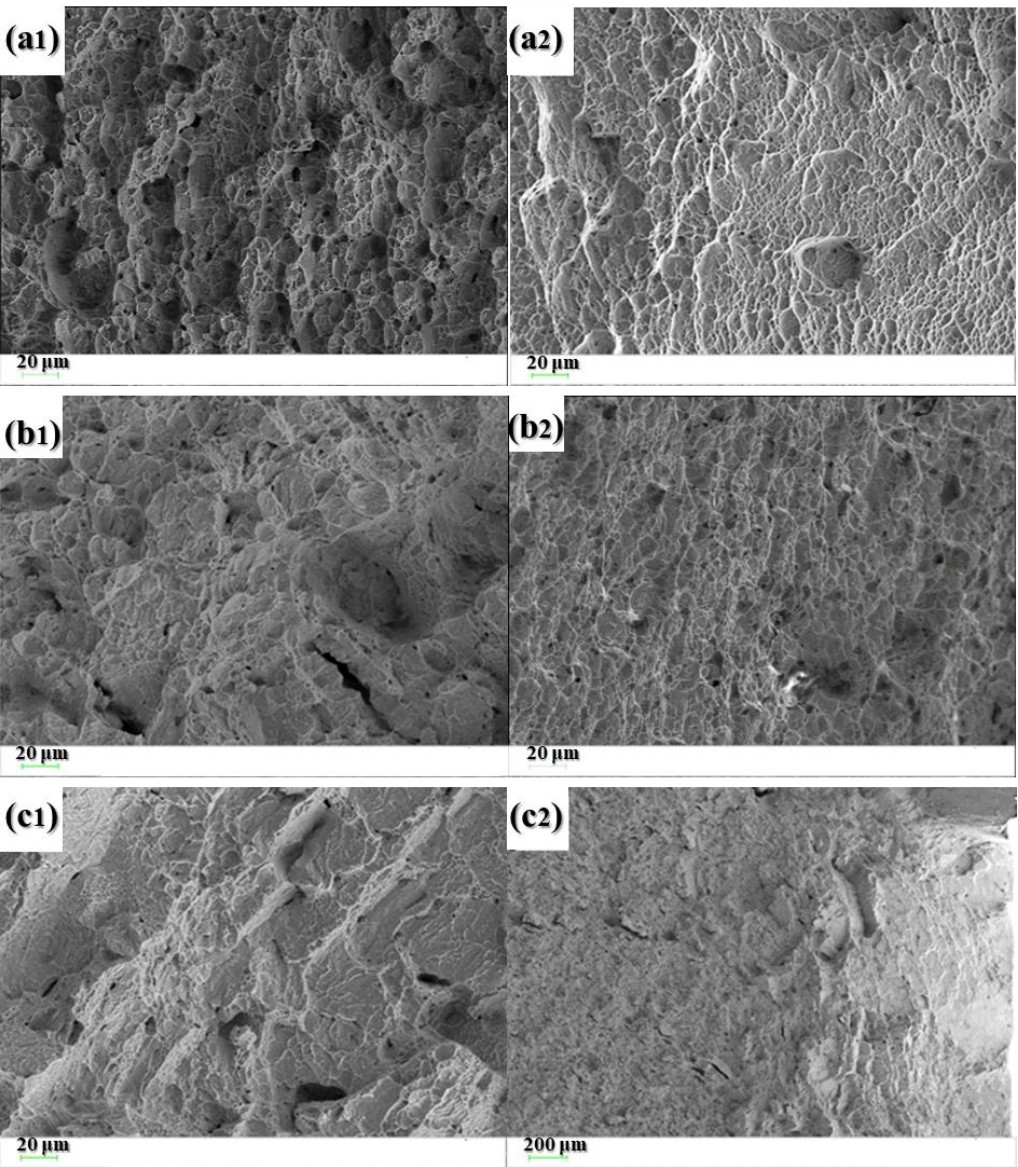

**Figure 13.** Impact fracture analysis of WM of TC4-welded-joint: (**a1**,**a2**) fibrous area; (**b1**,**b2**) radiation area; (**c1**,**c2**) shear lip area.

In Figure 13(b1), the radiative zone of the fracture surface shows dimples of varying sizes, with small dimples interweaving to form a net-like structure, while larger dimples are accompanied by cleavage planes. Micropores are clustered on the planar river-like pattern, with one larger micropore representing a tearing dimple. Near the shear lips, as shown in Figure 13(b2), the radiative zone of the fracture surface exhibits numerous small, deep, and densely packed dimples, characteristic of ductile fracture. Additionally, there are some cleavage planes distributed and relatively fewer deep micropores and large micropores.

The presence of second-phase inclusions within the micropores contributes to its relatively good toughness, which is associated with smaller grain size in the lower part of the WM.

Figure 13(c1) elucidates the detection of numerous significant cleavage steps and tear ridges present in the shear lips area, with dimples of differing scales dispersed across the cleavage terraces, primarily illustrating transgranular fracture phenomena. In Figure 13(c2), this area is mainly composed of river-like steps and numerous small cleavage steps, along with abundant tiny tear ridges. The specimen uniformly exhibits traits indicative of brittle fracture throughout. This phenomenon is primarily attributed to the persistent presence of $\alpha'$ martensite and Widmanstatten structures within the weld zone [38]. The coalescence of these microstructural features significantly elevates the brittleness of the matrix. Consequently, such brittle characteristics are prominently manifested during the terminal phase of impact-induced fracturing. Overall, the analysis indicates that, while maintaining welding quality, the impact toughness of the WM can be maintained at a relatively good level, closely approximating that of the BM [39].

In comparison to existing literature, our study demonstrates superior impact toughness in the WM and HAZ of titanium alloy weld joints. Although the impact performance of the weld zone has not exceeded that of the base material, it still demonstrates commendably good performance. Specifically, our findings indicate that the impact absorption energy shows enhanced toughness compared to the research by Balasubramanian et al. [36]. Furthermore, the study by Cui et al. [40] reveals that although the impact toughness of the weld metal does not surpass that of the base material, the presence of high-angle grain boundaries positively contributes to the impact toughness. This suggests that our welding process may offer optimized conditions that preserve or enhance joint performance without compromising material toughness.

### 3.5.2. Failure Mechanism of Impact Performance

To enhance the analysis of the microstructural factors contributing to impact crack failure, we employed TEM to examine the microstructure adjacent to the crack. We selected samples located near the impact fracture surface of the WM for detailed TEM characterization, as illustrated in Figure 14. The analysis of bright-field images and selected area electron diffraction (SAED) results revealed the presence of a small amount of β phase between the lamellar $\alpha/\alpha'$ phases, confirming the validity of the XRD and microscopic structure analyses. Characterization showed the existence of a small number of dislocations within some $\alpha/\alpha'$ phases, while the β phase was precisely located at the interfaces. When subjected to external impact loads, the main impediments to crack propagation are concentrated at the interfaces near the β phase, as these interfaces typically have higher strain capacities. Therefore, they can initially resist external loads, and when many such interfaces appear, they can hinder further crack propagation.

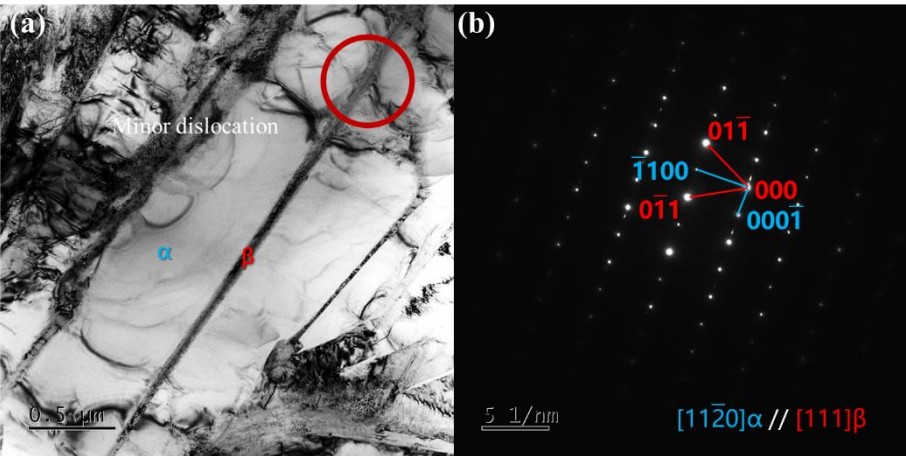

**Figure 14.** (**a**) TEM image of the microstructure near the fracture; (**b**) the diffraction spots.

Figure 15 illustrates a schematic diagram of a failure mechanism based on the results analysis. It is generally believed that during the impact process, cracks originate from micropores generated by plastic deformation due to external impact forces [41]. As the micropores expand, microscopic cracks penetrate columnar grains, mainly propagating along internal phase boundaries. When the angle of the phase boundary reaches the minimum energy for fracture, the crack continues to propagate. However, when the angle between the phase boundary and the crack propagation angle is small, the ability of crack propagation decreases, eventually leading to its cessation.

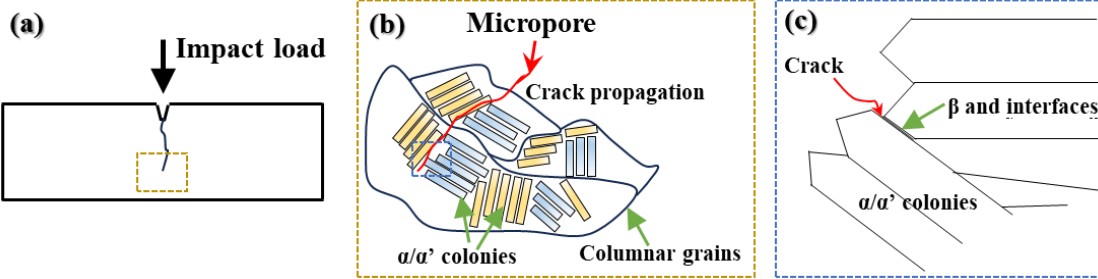

**Figure 15.** Schematic diagram of a failure mechanism of impact performance in WM of the joint: (**a**) impact diagram; (**b**) the enlarged area in (**a**); (**c**) the enlarged area in (**b**).

## 4. Conclusions

This study employed multi-layer multi-pass welding to complete the laser-MIG hybrid welding of 12 mm thick titanium alloy, and through performance testing and characterization, the following findings were made:

(1) Minor differences were observed in the microstructure among different passes, with all exhibiting $\alpha/\alpha'$ phases and a minor presence of β phases, predominantly in basket-weave structures, primarily consisting of needle-like $\alpha'$ martensite. However, it was noted that the upper regions tended to display Widmanstatten structures, which are less conducive to the enhancement of joint toughness.

(2) The upper layer of the 12 mm laser-MIG composite multi-layer multi-pass weld exhibited the highest hardness value, peaking at approximately 420 HV. This contrasted with the lower weld seam, which displayed a smaller hardness peak and variances in bending performance across different passes. Crucially, the toughness in the lower part of the joint weld seam outperformed that in the upper part, correlating with the refined grain size due to repetitive welding heat cycles, highlighting the intricate interplay between welding parameters and mechanical properties.

(3) The impact performance within the weld seam area did not reach the levels of the HAZ or the base metal, attaining 91.3% of the base metal's impact absorption capacity. This reduction in toughness, particularly in the upper section, can primarily be attributed to an increased presence of pores. Furthermore, the variation in grain size and phase content due to multi-layer multi-pass welding creates a differential in toughness across distinct areas of the fracture surface, illustrating the complex relationship between microstructure and mechanical properties in welded joints.

(4) Analysis of the microstructure near the failure surface allows us to delineate a failure mechanism: the initiation of cracks stems from the expansion of micropores during deformation. The existing β phase between the α phases and at their interfaces acts as an effective barrier against further expansion when the crack propagation angle remains small, elucidating the intricate fracture behavior of welded joints under stress.

**Author Contributions:** Conceptualization, J.X. and P.L.; methodology, W.F.; software, G.Z.; validation, P.L. and L.L.; formal analysis, W.F.; investigation, G.Z.; resources, J.X.; data curation, G.Z.; writing—original draft preparation, P.L.; writing—review and editing, J.X. All authors have read and agreed to the published version of the manuscript.

**Funding:** This research received to external funding.

**Institutional Review Board Statement:** There are no ethical issues with this study.

**Data Availability Statement:** The raw/processed data required to reproduce these findings cannot be shared at this time as the data also forms part of an ongoing study.

**Conflicts of Interest:** The authors declare that they have no known competing financial interests or personal relationships that could have appeared to influence the work reported in this paper.

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
