# Peer review of "Microstructure and Impact Toughness of Laser-Arc Hybrid Welded Joint of Medium-Thick TC4 Titanium Alloy"

_coatings, doi:10.3390/coatings14040395_

Round 1

Reviewer 1 Report

Comments and Suggestions for Authors

The authors investigated the welding process and its effects on the microstructure and impact toughness of TC4 titanium alloy. Specifically, they employed a laser-arc hybrid welding technique to join medium-thick sections of the TC4 titanium alloy. The focus was on understanding how this welding method influences the microstructural characteristics and the toughness properties of the welded joint. Through various analyses, including microstructural examination and impact toughness testing, the authors aimed to assess the weld quality and its suitability for practical applications in terms of mechanical performance and structural integrity. I think the study should be re-evaluated after major revision.

-Give brief information about the characterization methods used in the Abstract section.

-2.1. In the Material section, indicate where the materials were obtained.

-Figure 2a is not referenced in the text. Please include Figure 2a where you provide information about Figure 2a.

-According to what standards were the hardness, impact and bending tests performed? Add about standards with citing.

-3.1. Microstructure Analysis, 3.1.1 Microscopic Analysis and 3.1.2 XRD Analysis... So, XRD Analysis is a microstructure analysis? Please organize the headings well.

-Is Fig. 11 caption correct? The figure contains a macroscopic image, but the caption talks about the hardness distribution. Additionally, all images in Figure 11 should be described in the figure caption.

-All figures should be described in the text. For example, where is Figure 10 in the text? Please check this for all shapes. Cite Figure 10 in the text at the relevant place.

-I didn't see any discussion in this study? Are the results found by the authors consistent with similar results given in the literature? Either a separate discussion section should be included, or in the results section, the authors should compare the results they obtained (under all experimental result headings) with similar results obtained in the literature and emphasize that there is overlap.

-The Conclusion section should be equipped with expressions that emphasize the originality of the study.

Author Response

Detailed responses to the reviewers’ comments

Dear editor and reviewers:

Thanks for your positive suggestions on our ‘coatings-2915898’ entitled Microstructure and impact toughness of laser-arc hybrid welded joint of medium-thick TC4 titanium alloy”. We express our appreciation for reviewers’ critical comments and kind suggestions to improve this manuscript. We revised the manuscript in accordance with the reviewers’ comments, and carefully proof-read the manuscript to minimize typographical and grammatical errors. The point-by-point responses to the concerns have been attached below this page, which is given in normal font and changes/additions to the manuscript are given in yellow text with red font. Besides, all the changes are presented in the revised manuscript. 

We believe that the quality of this manuscript has been significantly improved by modifying it based on the reviewer comments. We sincerely hope that all the corrections will meet your requirements and expectations, and look forward to your approval on this publication.

Thank you for your time and consideration.

Yours Sincerely

Peng Luo 

Point-by-point responses

Responses to Reviewer #1:

The authors investigated the welding process and its effects on the microstructure and impact toughness of TC4 titanium alloy. Specifically, they employed a laser-arc hybrid welding technique to join medium-thick sections of the TC4 titanium alloy. The focus was on understanding how this welding method influences the microstructural characteristics and the toughness properties of the welded joint. Through various analyses, including microstructural examination and impact toughness testing, the authors aimed to assess the weld quality and its suitability for practical applications in terms of mechanical performance and structural integrity. I think the study should be re-evaluated after major revision.

  1. The reviewer’s comment: Give brief information about the characterization methods used in the Abstract section.
  • Response:We are very grateful to the reviewer’s useful comments and helpful suggestion. The brief information about the characterization methods has been added in the Abstract. The changes in the revised manuscript are marked in yellow with red font, which is listed below.
  • Abstract relative parts:Microstructural scrutiny, employing optical microscopy, SEM, and TEM, unveils a consistent composition across weld passes, with prevailing α/α’ phases interspersed with some β phase, resulting in basket-weave structures primarily dominated by acicular α’ martensite… (Page 1, Line 6-7)

  1. The reviewer’s comment: 2.1. In the Material section, indicate where the materials were obtained.
  • Response:Thanks for your helpful suggestion. Taking your suggestion into consideration, we have checked and amended the sentences, which are also listed below.
  • The base material (BM) used is TC4, also known as Ti-6Al-4V, which is one of the α+β titanium alloys. This material, with a thickness of 12 mm, was fabricated through the hot rolling process by Baoti Co. LTD, China.(Page 2, Line 20-22)

  1. The reviewer’s comment: Figure 2a is not referenced in the text. Please include Figure 2a where you provide information about Figure 2a.
  • Response:We appreciate your valuable advice. In light of your suggestion, we have reviewed and amended the sentences, which are outlined below.
  • The primary equipment utilized during the welding process is depicted in Figure 21. The laser-arc hybrid welding system, as shown in Figure 21(a), consists of a FANUC robot, an IPG Photonics YLS-10000 laser with a 10 kW maximum power, 0.6 mm spot diameter, 300 mm focal length, and a Fronius TPS4000 arc welder.As illustrated in Figure 21(b), the laser-MIG hybrid welding process for medium-thickness plates adopts a leading laser and trailing arc configuration, (Page 6, Line 7-12)

  1. The reviewer’s comment: According to what standards were the hardness, impact and bending tests performed? Add about standards with citing.
  • Response:Thank you for your helpful suggestion. Following your advice, we have examined and adjusted the sentences, as detailed below.
  • The schematic diagram of hardness testing positions is shown in Figure 4, utilizing an HVS-20 Vickers microhardness tester, with tests conducted in accordance with GB/T 4340.1.(Page 9, Line4-6)
  • Bend testing, a common method for assessing mechanical properties, is conducted to evaluate the plasticity and bonding strength between the weld and the BM, with tests carried out in accordance with GB/T 2653 standards.(Page 9, Line12-14)
  • The impact toughness testing equipment used is the PTM220 impact tester, with tests performed at least 3 times once a weld layer in accordance with GB/T 2650 standards.(Page 10, Line1-3)

  1. The reviewer’s comment: 3.1. Microstructure Analysis, 3.1.1 Microscopic Analysis and 3.1.2 XRD Analysis... So, XRD Analysis is a microstructure analysis? Please organize the headings well.
  • Response:We are very grateful to the reviewer’s useful comments and helpful suggestion. Actually, we admitted that XRD analysis is generally not considered a part of ‘microstructural analysis’. Although XRD can be utilized to study crystalline structures and lattice parameters, it predominantly focuses on the arrangement of atoms within crystals rather than providing direct observation of the material's microstructure, such as particles, grains, or grain boundaries. Hence, in our context of "Microstructure Analysis," XRD analysis has been typically listed separately and distinguished from microscopic analysis techniques, such as optical and electron microscopy. Following your advice, we have examined and adjusted the sentences, as detailed below.
  • 1.1. Microscopic characterization
  • 2.XRD analysis (Page 15, Line3)

  1. The reviewer’s comment: Is Fig. 11 caption correct? The figure contains a macroscopic image, but the caption talks about the hardness distribution. Additionally, all images in Figure 11 should be described in the figure caption.
  • Response:Thank you for your valuable suggestion. Upon considering your advice, we have reviewed and updated the caption for Fig. 11, as detailed below.
  • 1112Macroscopic fracture morphology of laser-MIG hybrid welded TC4 joint: (a1-a3) sampled WM as the center; (b1-b3) sampled HAZ as the center; (c) sampled BM as the center. (Page 20, Line4-5)

  1. The reviewer’s comment: All figures should be described in the text. For example, where is Figure 10 in the text? Please check this for all shapes. Cite Figure 10 in the text at the relevant place.
  • Response:Thank you for your valuable suggestion. Upon considering your advice, we have reviewed this for all shapes and cited Figure 10 in the text at the relevant place, as detailed below.
  • The hardness curve, as depicted in Figure 1011, exhibits a convex profile, indicating that the hardness in the vicinity of the weld bead's center is significantly higher than that in the BM.(Page 16, Line3-5)
  • The primary equipment utilized during the welding process is depicted in Figure 21. The laser-arc hybrid welding system, as shown in Figure 21(a), consists of a FANUC robot, an IPG Photonics YLS-10000 laser with a 10 kW maximum power, 0.6 mm spot diameter, 300 mm focal length, and a Fronius TPS4000 arc welder.As illustrated in Figure 21(b), the laser-MIG hybrid welding process for medium-thickness plates adopts a leading laser and trailing arc configuration, (Page 6, Line 7-12)

  1. The reviewer’s comment: I didn't see any discussion in this study? Are the results found by the authors consistent with similar results given in the literature? Either a separate discussion section should be included, or in the results section, the authors should compare the results they obtained (under all experimental result headings) with similar results obtained in the literature and emphasize that there is overlap.
  • Response:We appreciate your valuable advice. In fact, the discussion section of our work is primarily presented following the results in each subsection, specifically within the third part titled '3. Results and discussion'. Our findings share similarities with those reported in the literature, yet they also exhibit some of differences. Perhaps the perception of our analysis and discussion appearing unique stems from a lack of direct comparison with results from other literary works. Hence, as you suggested, we have added content to the discussion section to emphasize both the overlaps and the distinctions. The modifications are listed below.
  • These findings suggest that HAZdemonstrates enhanced toughness properties, likely due to the thermal cycling experienced during the welding process, which may refine the grain structure or alter the phase composition, thus improving the impact resistance. The BM, with an impact absorption energy of 33.5 J, shows the highest toughness, indicating that the original material properties are largely preserved outside of the direct influence of welding heat. Additionally, the variance in impact absorption energy, represented by the standard deviations, indicates a level of inconsistency within the zones, with the WM area showing a higher variability (standard deviation of 2.18) compared to the HAZ (standard deviation of 1.90). This variation could be attributed to factors such as welding speed, temperature gradients, and material flow during the welding process, which can introduce heterogeneities in the microstructure, particularly in the WM. The difference between the WM and HAZ impact toughness also highlights the right combination can minimize the adverse effects of welding on the microstructure and enhance the overall performance of the welded joint [35-36]. (Page 19, Line 9- Page 20, Line 1)
  • Figure 13(c1) elucidates the detection of numerous significant cleavage steps and tear ridges present in the shear lips area, with dimples of differing scales dispersed across the cleavage terraces, primarily illustrating trans granular fracture phenomena. In Figure 13(c2), this area is mainly composed of river-like steps and numerous small cleavage steps, along with abundant tiny tear ridges. The specimen uniformly exhibits traits indicative of brittle fracture throughout. This phenomenon is primarily attributed to the persistent presence of α' martensite and Widmanstatten structures within the weld zone [39]. The coalescence of these microstructural features significantly elevates the brittleness of the matrix. Consequently, such brittle characteristics are prominently manifested during the terminal phase of impact-induced fracturing. Overall, the analysis indicates that, while maintaining welding quality, the impact toughness of the WM can be maintained at a relatively good level, closely approximating that of the BM [40]. (Page23, Line7-19)
  • In comparison to existing literature, our study demonstrates superior impact toughness in the WM and HAZ of titanium alloy weld joints. Although the impact performance of the weld zone has not exceeded that of the base material, it still demonstrates commendably good performance. Specifically, our findings indicate that the impact absorption energy shows enhanced toughness compared to the research by Balasubramanian et al. [36]. Furthermore, the study by Cui et al. [41] reveals that although the impact toughness of the weld metal does not surpass that of the base material, the presence of high-angle grain boundaries positively contributes to the impact toughness. This suggests that our welding process may offer optimized conditions that preserve or enhance joint performance without compromising material toughness.(Page 23, Line 20- Page 24, Line 8)
  • The reference added below:

[35] doi.org/10.1016/j.pnsc.2024.03.001;

[36] doi.org/10.1016/S1003-6326(11)60850-9;

[39] doi.org/10.1108/GS-11-2019-0052;

[40] doi.org/10.3390/ma16041509;

[41] doi.org/10.1016/j.jmapro.2019.01.031.

  1. The reviewer’s comment: The Conclusion section should be equipped with expressions that emphasize the originality of the study.
  • Response:We appreciate your valuable advice. Taking your suggestion into consideration, we have checked and amended the sentences of the conclusion, which are also listed below.
  • Conclusions:
  • (1) Minor differences were observed in the microstructure among different passes, with all exhibiting α/α’ phases and a minor presence of β phases, predominantly in basket-weave structures, primarily consisting of needle-like α’ martensite. However, it was noted that the upper regions tended to display Widmanstatten structures, which are less conducive to the enhancement of joint toughness.
  • (2) The upper layer of the 12 mm laser-MIG composite multi-layer multi-pass weld exhibited the highest hardness value, peaking at approximately 420 HV. This contrasted with the lower weld seam, which displayed a smaller hardness peak and variances in bending performance across different passes. Crucially, the toughness in the lower part of the joint weld seam outperformed that in the upper part, correlating with the refined grain size due to repetitive welding heat cycles, highlighting the intricate interplay between welding parameters and mechanical properties.
  • (3) The impact performance within the weld seam area did not reach the levels of the HAZ or the base metal, attaining 91.3% of the base metal’s impact absorption capacity. This reduction in toughness, particularly in the upper section, can primarily be attributed to an increased presence of pores. Furthermore, the variation in grain size and phase content due to multi-layer multi-pass welding creates a differential in toughness across distinct areas of the fracture surface, illustrating the complex relationship between microstructure and mechanical properties in welded joints.
  • (4) Analysis of the microstructure near the failure surface allows us to delineate a failure mechanism: the initiation of cracks stems from the expansion of micropores during deformation. The existing β phase between the α phases and at their interfaces acts as an effective barrier against further expansion when the crack propagation angle remains small, elucidating the intricate fracture behavior of welded joints under stress.

Reviewer 2 Report

Comments and Suggestions for Authors

The study titled "Microstructure and impact toughness of laser-arc hybrid welded joint of medium-thick TC4 titanium alloy" is on the impact properties of the joint of TC4 titanium alloy with laser arc (MIG) hybrid welding method. In the study, microstructure examination, hardness and bending tests, and toughness properties were examined according to welding passes. In their study, the authors state the feasibility of laser-MIG hybrid welding and the reliability of mechanical properties for medium-thickness TC4 alloy sheets. In its current state, the study contains interesting and innovative results. However, answers are required to some of the questions listed below.

1.      It should be added in line 109 that the Ti6Al4V alloy belongs to the TC4 class.

2.      Are the chemical composition values in Table 1 catalog values and how were they obtained by the researchers? Reference should be added if necessary. Likewise, how were the mechanical property values in Table 2 obtained?

3.      Preparation details of the microstructures in Figure 1 are missing. Also, why were both optical and SEM images given? The devices used to obtain it should be added. From which direction of the plate were microstructures obtained? Transversely or longitudinally? Microstructure preparation is given in the "2.3.1. Microstructure characterization" section. However, it would have been better to move these raw microstructures to the results and discussion section.

4.      It would be better if a schematic illustration was written instead of a schematic diagram in Figure 2. Additionally, the difference between Figure 2 a and b should have been clearly stated in the text.

5.      In line 154, technical details of TEM sample preparation are missing.

6.      XRD device brand and scanning speed should be added.

7.      Figure 4 is not a diagram, but a schematic representation.

8.      Bending and impact tests must be carried out with technical details and international standard numbers. It would be appropriate to give the bending test sample with its standard dimensions, such as the impact test schematic representation.

9.      It would be more appropriate if the 1st, 2nd and 3rd welds in Figure 6 were given as welding passes instead. Also, do these layers have clear boundaries? Has deep etching been performed? Figure 6 caption (a) welded joint is not suitable. instead, the weld joint microstructure distribution can be written.

10.  (a)-(f) should be stated in detail in the Figure 8 caption. SEM was not used in microstructure images. It would be appropriate if α, α', β phases could be specified on microstructure images.

11.  In Figure 10, instead of upper, middle, lower layer, 1.2.3. The pass must be spread. In addition, these expressions should be written with the same expression everywhere in the text. In some places, as in line 283, cover pass weld and in some places cometic weld layer expressions make it difficult to read. Please use systematically consistent expressions.

12.  Check the caption of Figure 5. The dimensions of the sample should be added in parentheses (mm).

13.  The term "face bending" in line 291 conflicts with the terms "positive bending" in table 4. A standard common term should be used. Additionally, technical details of the bending test modes are missing in the experimental studies section.

14.  Standard deviation values of the average of bending test and impact test results should also be given. Information about the number of repeat tests performed should be included.

15.  The microstructure and literature-supported discussion section on bending test and impact test results is weak and should be expanded.

16.  It was not appropriate to discuss failure analysis with TEM in the "3.4.2. Failure mechanism of impact performance" section. It would be appropriate to include TEM and SAED analyzes in microstructure characterization. For failure analysis, cross-sectional examination of the fractured surfaces would be sufficient.

Author Response

Detailed responses to the reviewers’ comments

Dear editor and reviewers:

Thanks for your positive suggestions on our ‘coatings-2915898’ entitled Microstructure and impact toughness of laser-arc hybrid welded joint of medium-thick TC4 titanium alloy”. We express our appreciation for reviewers’ critical comments and kind suggestions to improve this manuscript. We revised the manuscript in accordance with the reviewers’ comments, and carefully proof-read the manuscript to minimize typographical and grammatical errors. The point-by-point responses to the concerns have been attached below this page, which is given in normal font and changes/additions to the manuscript are given in yellow text with red font. Besides, all the changes are presented in the revised manuscript. 

We believe that the quality of this manuscript has been significantly improved by modifying it based on the reviewer comments. We sincerely hope that all the corrections will meet your requirements and expectations, and look forward to your approval on this publication.

Thank you for your time and consideration.

Yours Sincerely

Peng Luo 

Responses to Reviewer #2:

The study titled "Microstructure and impact toughness of laser-arc hybrid welded joint of medium-thick TC4 titanium alloy" is on the impact properties of the joint of TC4 titanium alloy with laser arc (MIG) hybrid welding method. In the study, microstructure examination, hardness and bending tests, and toughness properties were examined according to welding passes. In their study, the authors state the feasibility of laser-MIG hybrid welding and the reliability of mechanical properties for medium-thickness TC4 alloy sheets. In its current state, the study contains interesting and innovative results. However, answers are required to some of the questions listed below.

  1. The reviewer’s comment: It should be added in line 109 that the Ti6Al4V alloy belongs to the TC4 class.
  • Response:Thanks for your helpful suggestion. Taking your suggestion into consideration, we have checked and amended the sentences, which are also listed below.
  • The base material (BM) used is TC4, also known as Ti-6Al-4V, which is one of the α+β titanium alloys. This material, with a thickness of 12 mm, was fabricated through the hot rolling process by Baoti Co. LTD, China.(Page 5, Line 20-22)

  1. The reviewer’s comment: Are the chemical composition values in Table 1 catalog values and how were they obtained by the researchers? Reference should be added if necessary. Likewise, how were the mechanical property values in Table 2 obtained?
  • Response:We appreciate your valuable advice. The chemical composition listed in Table 1 is provided by the manufacturer, and the testing methodologies are interpreted by the manufacturer due to confidentiality clauses in the commercial contract; details cannot be disclosed. However, TC4 is a common titanium alloy, and its production and alloy composition adhere to certain standards. Any composition that meets these standards is deemed acceptable. These standards comply with GB/T 3620.1. Similarly, the mechanical properties of the base material and welding wire are also assessed according to specific testing standards, which, as understood, align with GB/T 16957 and GB/T 228.1 as per the manufacturer's testing. Given that the primary focus of this paper is on the organizational performance of welding joints, these sections are introduced briefly, allocating more space to welding-related analyses. Taking your suggestion into consideration, we have checked and amended the relevant sentences, which are also listed below.
  • The base material (BM) used is TC4, also known as Ti-6Al-4V, which is one of the α+β titanium alloys. This material, with a thickness of 12 mm, was fabricated through the hot rolling process by Baoti Co. LTD, China.(Page 5, Line 20-22)

  1. The reviewer’s comment: Preparation details of the microstructures in Figure 1 are missing. Also, why were both optical and SEM images given? The devices used to obtain it should be added. From which direction of the plate were microstructures obtained? Transversely or longitudinally? Microstructure preparation is given in the "2.3.1. Microstructure characterization" section. However, it would have been better to move these raw microstructures to the results and discussion section.
  • Response:Thank you for your valuable suggestion. We have added the schematic diagram of the sampling position. The details prepared in the original Figure 1 are the same as those in 2.3.1 except for the different sampling positions, so there is no need to repeat them. At the same time, the image is given in order to enlarge the size of the microstructure and to better compare with the microstructure at the joint site. The device to obtain it is given in 2.3.1. It is observed longitudinally from the rolling direction perpendicular to the plate, and this part has been added and improved in the article. Upon considering your advice, we have reviewed and moved this section to a later section, as detailed below.
  • The overview diagram of sampling position of weld properties is shown in Figure 3.Each pair of plates should be tested in the middle of the weld as far as possible. (Page 7, Line15-16)
  •  
  • 3 Schematic diagram of sampling position.(Page 8, Line1-2)
  • the microstructure of the joint and BM samples requires characterization and analysis.(Page 8, Line5-6)
  • …using a Zeiss optical microscope (OM, ZEISS temi), a scanning electron microscope (SEM, ZEISS Gemini 300), …(Page 8, Line9-11)
  • The microstructure of the BM is illustrated in Figure 6, and it is observed longitudinally from the rolling direction perpendicular to the plate,… (Page 10, Line12-13)
  • 1 6Microstructure of TC4 base metal: (a) OM; (b) SEM. (Page 10, Line17)

  1. The reviewer’s comment: It would be better if a schematic illustration was written instead of a schematic diagram in Figure 2. Additionally, the difference between Figure 2 a and b should have been clearly stated in the text.
  • Response: We appreciate your valuable advice. Considering your suggestion, we have reviewed and revised the sentences. However, from our perspective, incorporating a schematic diagram would provide a more intuitive understanding.In addition, the (original) Figure 2a is the schematic diagram of the welding system, and Figure 2b is some details of the welding process, which have been modified in the paper, as detailed below.
  • The primary equipment utilized during the welding process is depicted in Figure 21. The laser-arc hybrid welding system, as shown in Figure 21(a), consists of a FANUC robot, an IPG Photonics YLS-10000 laser with a 10 kW maximum power, 0.6 mm spot diameter, 300 mm focal length, and a Fronius TPS4000 arc welder.As illustrated in Figure 21(b), the laser-MIG hybrid welding process for medium-thickness plates adopts a leading laser and trailing arc configuration, (Page 6, Line 7-12)

  1. The reviewer’s comment: In line 154, technical details of TEM sample preparation are missing.
  • Response:Thanks for your helpful suggestion. Taking your suggestion into consideration, we have checked and amended the sentences, which are also listed below.
  • Among them, the TEM samples were prepared using an EM Precision Cutter (Model EM-PC300) to slice ultrathin sections. These sections were then thinned to electron transparency using an ion milling machine (Model IM4000). The final thickness was monitored under a light microscope to ensure optimal electron transmittance for TEM analysis.(Page 8, Line12-16)

  1. The reviewer’s comment: XRD device brand and scanning speed should be added.
  • Response:Thanks for your helpful suggestion. Taking your suggestion into consideration, we have checked and amended the sentences, which are also listed below.
  • The XRD analysis was conducted using a Panalytical Empyrean diffractometer, equipped with a Cu Kα radiation source. The scanning speed was set to 1°/min, covering a 2θ range from 30° to 80°. This setup ensures comprehensive coverage and detailed data collection for accurate phase identification and lattice parameter calculations.(Page 8, Line18-Page 9, Line2)

  1. The reviewer’s comment: Figure 4 is not a diagram, but a schematic representation.
  • Response:Thanks for your helpful suggestion. Taking your suggestion into consideration, we have amended the sentences, which are also listed below.
  • 5Schematic representation of the position of hardness tests. (Page 9, Line10-11)

  1. The reviewer’s comment: Bending and impact tests must be carried out with technical details and international standard numbers. It would be appropriate to give the bending test sample with its standard dimensions, such as the impact test schematic representation.
  • Response:Thanks for your helpful suggestion. Test details and standard have been added in the article. However, due to the bending sample is simple and the bending test and the impact test are very common, there is no need to draw more details, and bending test sample of 200×20×10 mm is written. Taking your suggestion into consideration, we have amended the sentences, which are also listed below.
  • Bend testing, a common method for assessing mechanical properties, is conducted to evaluate the plasticity and bonding strength between the weld and the BM, with tests carried out in accordance with GB/T 2653 standards.(Page 9, Line12-14)
  • The three-point bending apparatus, with a bending core diameter of approximately 40mm and capable of bending angles up to 180°, is used to apply a continuous force at the midpoint between the supports until the specimen bends to the specified angle or exhibits visible cracking.(Page 9, Line16-20)
  • The impact toughness testing equipment used is the PTM220 impact tester, with tests performed at least 3 times once a weld layer in accordance with GB/T 2650 standards.(Page 10, Line1-3)

  1. The reviewer’s comment: It would be more appropriate if the 1st, 2nd and 3rd welds in Figure 6 were given as welding passes instead. Also, do these layers have clear boundaries? Has deep etching been performed? Figure 6 caption (a) welded joint is not suitable. instead, the weld joint microstructure distribution can be written.
  • Response:Thank you for your constructive suggestion. As you suggested, providing the weld metallography for the 1st, 2nd, and 3rd layers in the figure would indeed be appropriate. However, upon thorough investigation and comparison, it has been established that similar work has already been conducted by other scholars, as listed below. Their findings indicate that presenting these layers separately is unnecessary, as distinct boundaries between different levels are evident, and our final weld composition necessitates three layers rather than one or two. Consequently, from the perspective of subsequent performance analysis, this detail is not crucial. Nevertheless, we sincerely appreciate your suggestion. Considering your feedback, we have reviewed and revised the sentences, which are outlined below.
  • 67 (a)The microstructure distribution of the welded joint; (Page 12, Line2-3)
  • References mentioned below:

doi.org/10.1016/j.optlastec.2024.110569;

doi.org/10.3390/cryst12070977;

doi.org/10.1007/s00170-016-8926-4.

  1. The reviewer’s comment: (a)-(f) should be stated in detail in the Figure 8 caption. SEM was not used in microstructure images. It would be appropriate if α, α', β phases could be specified on microstructure images.
  • Response:Thanks for your helpful suggestion. Due to the distinct microstructure of the weld zone, phase distribution across different grains can be effectively compared and distinguished using an optical microscope. However, under high magnification with SEM, comparing the distribution and morphology of different phases becomes challenging. Hence, we opt to utilize the optical microscope for analytical purposes, while SEM images highlighting prominent features have already been provided above. Taking your suggestion into consideration, we have amended the sentences, which are also listed below.
  • 89Microstructure of the different zone of laser-MIG hybrid welded TC4 joint. (a)WM in 3rd# weld; (b) HAZ in 3rd# weld; (c) WM in 2nd# weld; (d) HAZ in 2nd# weld; (e) WM in 1st# weld; (f) HAZ in 1st# weld. (Page 14, Line2-5)

  1. The reviewer’s comment: In Figure 10, instead of upper, middle, lower layer, 1.2.3. The pass must be spread. In addition, these expressions should be written with the same expression everywhere in the text. In some places, as in line 283, cover pass weld and in some places cometic weld layer expressions make it difficult to read. Please use systematically consistent expressions.
  • Response:We appreciate your valuable feedback. In response to your comments, we have performed additional testing and independently generated new illustrations. The revised results are now presented in the subsequent figure. Following your recommendation, we have also reviewed and corrected the text, as delineated below.
  • As is depicted, Figure 11(a)(b)(c)show the horizontal hardness test results of the three layers of the weld, respectively. (Page 16, Line5-6)
  • Vertically, the hardness value is highest in the cosmetic weld(3rd#), … (Page 16, Line16-17)
  • The increased hardness in WM of titanium alloy welding is primarily due to the microstructure formed by rapid cooling during the welding process. In the WM, rapid cooling can lead to the formation of fine, hardened α’martensitic phases. This fine martensitic structure is much harder than the original structure of the titanium alloy, thus increasing the overall hardness of the welded area[30]. This is also related to the observed distribution of Widmanstatten structures [31-32]. (Page 16, Line17- Page 17, Line1)
  •  
  • 11Hardness distribution of laser-MIG hybrid welded TC4 alloy joint:(a)3rd# layer, (b)2nd# layer, (c)1st# layer. (Page 17, Line5-6)
  • Thereference added below:
  • [30]org/10.1016/j.msea.2010.10.070

[31] doi.org/10.1016/j.matchar.2021.111162

[32] doi.org/10.1016/j.jmst.2018.03.012

  1. The reviewer’s comment: Check the caption of Figure 5. The dimensions of the sample should be added in parentheses (mm).
  • Response:We appreciate your valuable advice. Taking your suggestion into consideration, we have checked and amended the sentences, which are also listed below.
  • (b) size of specimens(mm). (Page10, Line9)

  1. The reviewer’s comment: The term "face bending" in line 291 conflicts with the terms "positive bending" in table 4. A standard common term should be used. Additionally, technical details of the bending test modes are missing in the experimental studies section.
  • Response:We appreciate your valuable advice. The technical details of the bending test modes have been added. Taking your suggestion into consideration, we have checked and amended the sentences of the conclusion, which are also listed below.
  • Mode: face bending (Table 4)
  • Bend testing, a common method for assessing mechanical properties, is conducted to evaluate the plasticity and bonding strength between the weld and the BM, with tests carried out in accordance with GB/T 2653 standards.Rectangular bend specimens measuring 200×20×10 mm is cut from the test plates, and the specimens are fixed during testing. The three-point bending apparatus, with a bending core diameter of approximately 40mm and capable of bending angles up to 180°, is used to apply a continuous force at the midpoint between the supports until the specimen bends to the specified angle or exhibits visible cracking. (Page 9, Line12-19)

  1. The reviewer’s comment: Standard deviation values of the average of bending test and impact test results should also be given. Information about the number of repeat tests performed should be included.
  • Response:We appreciate your valuable advice. Taking your suggestion into consideration, we have added results and amended the sentences, which are also listed below.
  • The same parameter should be tested at least 3 times, and the corresponding data should be processed.(Page 9, Line20)
  • The impact toughness testing equipment used is the PTM220 impact tester, with tests performed at least 3 times once a weld layer in accordance with GB/T 2650 standards.(Page 10, Line1-3)
  • Particularly in the third group of specimens, the test force in the root bend test reaches41kN, which is slightly higher than the highest value of 30.10kN in the positive bend test. The standard deviations of the test forces for positive bending and back bending are 0.08 and 0.69, respectively. (Page 18, Line1-4) 
  • Table 4. Results of bending test.

Number

Mode

Bending angle (°)

Bending force (kN)

Average force (kN)

Standard Deviation

1-1

Face bending

9.4

30.05

30.02

0.08

2-1

14.8

29.90

3-1

15.7

30.10

1-2

Root bending

9.6

31.41

30.96

0.69

2-2

13.2

31.49

3-2

25.1

29.98

  • Table 5. Results of impact toughness test in different locations. (Unit of Akv: J)

Number

WM

HAZ

BM

1

33.1

34.0

34.0

2

31.0

30.7

33.0

3

27.8

35.2

33.5

Average (J)

30.6

33.3

33.5

Standard Deviation

2.18

1.90

0.41

  1. The reviewer’s comment: The microstructure and literature-supported discussion section on bending test and impact test results is weak and should be expanded.
  • Response:We appreciate your valuable advice. Taking your suggestion into consideration, we have revised and amended the sentences, which are also listed below.
  • This disparity in bending performance can be correlated to the microstructural characteristics observed within the  WM. Specifically, the superior performance of the 3rd# weld bead suggests a finer and more homogeneous microstructure, which is known to enhance the toughness and load-bearing capacity of the material under stress. Furthermore, the observed standard deviations imply a more consistent mechanical behavior in the face bend tests compared to the root bend tests, potentially indicating a more uniform distribution of residual stresses and microstructural features across the tested zones. Future studies could further elucidate these relationships by correlating microstructural features with mechanical properties across different sections of the weld joint, as has been suggested in the literature [33-34]. This comprehensive approach can offer deeper insights into the influence of welding parameters on joint performance.(Page 18, Line10-20)
  • In comparison to existing literature, our study demonstrates superior impact toughness in the WM and HAZof titanium alloy weld joints. Although the impact performance of the weld zone has not exceeded that of the base material, it still demonstrates commendably good performance. Specifically, our findings indicate that the impact absorption energy shows enhanced toughness compared to the research by Balasubramanian et al. [36]. Furthermore, the study by Cui et al. [41] reveals that although the impact toughness of the weld metal does not surpass that of the base material, the presence of high-angle grain boundaries positively contributes to the impact toughness. This suggests that our welding process may offer optimized conditions that preserve or enhance joint performance without compromising material toughness. (Page 23, Line20- Page 24, Line8)
  • The references added below:

[33] doi.org/10.3390/met12050873; 

[34] doi.org/10.2351/1.5089875; 

[36] doi.org/10.1016/S1003-6326(11)60850-9; 

[41] doi.org/10.1016/j.jmapro.2019.01.031.

  1. The reviewer’s comment: It was not appropriate to discuss failure analysis with TEM in the "3.4.2. Failure mechanism of impact performance" section. It would be appropriate to include TEM and SAED analyzes in microstructure characterization. For failure analysis, cross-sectional examination of the fractured surfaces would be sufficient.
  • Response:Thanks for your valuable However, I think you may not have read our analysis carefully. The TEM analysis here is not only for the microstructure characterization, but for the characterization of the microstructure near the longitudinal crack distribution. Its purpose is to further analyze the impact and characteristics of the impact crack at the microscopic scale from the perspective of the microstructure. Many scholars have made characterization analysis of the tissue near the crack and found many interesting conclusions. For example, DOI:10.1016/j.msea.2024.146144, DOI:10.1016/j.matchar.2022.112606 etc. Of course, we did not ignore the cross-sectional characterization analysis, which can be more comprehensive analysis. Taking your suggestion into consideration, we have checked and amended the sentences of the relevant sentences, which are also listed below.
  • To enhance the analysis of the microstructural factors contributing to impact crack failure, we employed TEM to examine the microstructure adjacent to the crack. We selected samples located near the impact fracture surface of the WM for detailed TEM characterization, as illustrated in Figure 14.(Page 24, Line10-13)
  • The reference mentioned below:

doi.org/10.1016/j.msea.2024.146144

doi.org/10.1016/j.matchar.2022.112606

Reviewer 3 Report

Comments and Suggestions for Authors

The manuscript needs modification for further consideration.

How can authors define what thickness range is called medium thick or thick? Any reference authors can provide.

The conclusion section needs to be more qualitative.

The research gap is missing in the manuscript and needs to be addressed properly with the author's contributions.

Few literature would be helpful in addressing the literature section and can be added by the authors, such as doi.org/10.1016/j.optlastec.2019.04.004; doi.org/10.1007/s40430-021-03294-w

Fracture analysis needs to be discussed in a more elaborative way. The joint appears to have a brittle structure.

Fig. 11. shows pores, the authors need to do an image analysis to come to this conclusion.

Fig. 10. Why is hardness higher at the fusion zone? Can the author relate this to microstructure?

Author Response

Detailed responses to the reviewers’ comments

Dear editor and reviewers:

Thanks for your positive suggestions on our ‘coatings-2915898’ entitled Microstructure and impact toughness of laser-arc hybrid welded joint of medium-thick TC4 titanium alloy”. We express our appreciation for reviewers’ critical comments and kind suggestions to improve this manuscript. We revised the manuscript in accordance with the reviewers’ comments, and carefully proof-read the manuscript to minimize typographical and grammatical errors. The point-by-point responses to the concerns have been attached below this page, which is given in normal font and changes/additions to the manuscript are given in yellow text with red font. Besides, all the changes are presented in the revised manuscript. 

We believe that the quality of this manuscript has been significantly improved by modifying it based on the reviewer comments. We sincerely hope that all the corrections will meet your requirements and expectations, and look forward to your approval on this publication.

Thank you for your time and consideration.

Yours Sincerely

Peng Luo 

Responses to Reviewer #3:

The manuscript needs modification for further consideration.

  1. The reviewer’s comment: How can authors define what thickness range is called medium thick or thick? Any reference authors can provide.
  • Response:We appreciate your valuable advice. Typically, in industrial contexts, steel plates with thicknesses ranging from 6 to 25 mm are commonly referred to as medium thick plates, a nomenclature that reflects industry standards. This terminology is corroborated by numerous studies within this thickness range, for example: DOI: 10.1016/j.jmapro.2019.07.035; DOI: 10.1007/s11665-021-06534-1; DOI: 10.1016/j.optlastec.2024.110569; DOI: 10.1007/s00170-022-09089-0, among others. Furthermore, our classification aligns with international standards such as ASTM A36/A36M-14. Thus, categorizing our 12 mm thick plate as a medium thick plate is consistent with both industry convention and scholarly research. Taking your suggestion into consideration, we have checked and amended the relevant
  • The references mentioned below:
  • org/10.1016/j.jmapro.2019.07.035;

doi.org/10.1007/s11665-021-06534-1;

doi.org/10.1016/j.optlastec.2024.110569;

doi.org/10.1007/s00170-022-09089-0.

  1. The reviewer’s comment:The conclusion section needs to be more qualitative.
  • Response:We appreciate your valuable advice. Taking your suggestion into consideration, we have checked and amended the sentences of the conclusion, which are also listed below.
  • Conclusions:
  • (1) Minor differences were observed in the microstructure among different passes, with all exhibiting α/α’ phases and a minor presence of β phases, predominantly in basket-weave structures, primarily consisting of needle-like α’ martensite. However, it was noted that the upper regions tended to display Widmanstatten structures, which are less conducive to the enhancement of joint toughness.
  • (2) The upper layer of the 12 mm laser-MIG composite multi-layer multi-pass weld exhibited the highest hardness value, peaking at approximately 420 HV. This contrasted with the lower weld seam, which displayed a smaller hardness peak and variances in bending performance across different passes. Crucially, the toughness in the lower part of the joint weld seam outperformed that in the upper part, correlating with the refined grain size due to repetitive welding heat cycles, highlighting the intricate interplay between welding parameters and mechanical properties.
  • (3) The impact performance within the weld seam area did not reach the levels of the HAZ or the base metal, attaining 91.3% of the base metal’s impact absorption capacity. This reduction in toughness, particularly in the upper section, can primarily be attributed to an increased presence of pores. Furthermore, the variation in grain size and phase content due to multi-layer multi-pass welding creates a differential in toughness across distinct areas of the fracture surface, illustrating the complex relationship between microstructure and mechanical properties in welded joints.
  • (4) Analysis of the microstructure near the failure surface allows us to delineate a failure mechanism: the initiation of cracks stems from the expansion of micropores during deformation. The existing β phase between the α phases and at their interfaces acts as an effective barrier against further expansion when the crack propagation angle remains small, elucidating the intricate fracture behavior of welded joints under stress.

  1. The reviewer’s comment: The research gap is missing in the manuscript and needs to be addressed properly with the author's contributions.
  • Response:Thanks for your helpful suggestion. Taking your suggestion into consideration, we have checked and amended the sentences, which are also listed below.
  • …, as has been suggested in the literature [33-34]. This comprehensive approach can offer deeper insights into the influence of welding parameters on joint performance.(Page 18, Line19-20)
  • In comparison to existing literature, our study demonstrates superior impact toughness in the WM and HAZof titanium alloy weld joints. Although the impact performance of the weld zone has not exceeded that of the base material, it still demonstrates commendably good performance. Specifically, our findings indicate that the impact absorption energy shows enhanced toughness compared to the research by Balasubramanian et al. [36]. Furthermore, the study by Cui et al. [41] reveals that although the impact toughness of the weld metal does not surpass that of the base material, the presence of high-angle grain boundaries positively contributes to the impact toughness. This suggests that our welding process may offer optimized conditions that preserve or enhance joint performance without compromising material toughness. (Page 23, Line20- Page 24, Line8)
  • The referencesadded below:

[33] doi.org/10.3390/met12050873; 

[34] doi.org/10.2351/1.5089875; 

[36] doi.org/10.1016/S1003-6326(11)60850-9; 

[41] doi.org/10.1016/j.jmapro.2019.01.031.

  1. The reviewer’s comment: Few literature would be helpful in addressing the literature section and can be added by the authors, such as doi.org/10.1016/j.optlastec.2019.04.004; doi.org/10.1007/s40430-021-03294-w
  • Response:We greatly appreciate your insightful suggestion. In response, we have incorporated the referenced documents into the pertinent sections of our manuscript, including the two previously mentioned. A detailed list of these amendments is provided below for your review.
  • Among these techniques, laser beam welding has emerged as a promising method for joining titanium alloys due to its high energy density, minimal heat input, and precise control over the welding process [14-15].Furthermore, several researchers have explored the integration of artificial intelligence algorithms and alternative strategies to enhance the monitoring and adjustment of the welding process [16-17]; yet, challenges persist in the welding of titanium alloys, specifically concerning these process issues. To mitigate these constraints, innovative hybrid welding techniques, merging laser welding with supplementary welding methods, have been introduced. Notably, the laser-MIG hybrid welding technique has emerged as a significant advancement, offering enhanced penetration depth, increased weld width, and improved modulation of weld microstructure [18]. (Page2, Line20-Page3, Line8)
  • Thereferences added below:

[15] doi.org/10.1016/j.optlastec.2019.04.004

[16] doi.org/10.1016/j.jmapro.2023.08.056

[17] doi.org/10.1007/s40430-021-03294-w

[18] doi.org/10.1007/s00170-016-8544-1

  1. The reviewer’s comment: Fracture analysis needs to be discussed in a more elaborative way. The joint appears to have a brittle structure.
  • Response:Thanks for your valuable As you said, there are some brittle structures in the shear lip region of the impact fracture, for which we have added detailed analysis. Taking your suggestion into consideration, we have checked and amended the sentences, which are also listed below.
  • Figure 13(c1) elucidates the detection of numerous significant cleavage steps and tear ridges present in the shear lips area, with dimples of differing scales dispersed across the cleavage terraces, primarily illustrating trans granular fracture phenomena. In Figure 13(c2), this area is mainly composed of river-like steps and numerous small cleavage steps, along with abundant tiny tear ridges. The specimen uniformly exhibits traits indicative of brittle fracture throughout. This phenomenon is primarily attributed to the persistent presence of α' martensite and Widmanstatten structures within the weld zone [39]. The coalescence of these microstructural features significantly elevates the brittleness of the matrix. Consequently, such brittle characteristics are prominently manifested during the terminal phase of impact-induced fracturing. Overall, the analysis indicates that, while maintaining welding quality, the impact toughness of the WM can be maintained at a relatively good level, closely approximating that of the BM [40]. (Page23, Line7-19)
  • In comparison to existing literature, our study demonstrates superior impact toughness in the WM and HAZof titanium alloy weld joints. Although the impact performance of the weld zone has not exceeded that of the base material, it still demonstrates commendably good performance. Specifically, our findings indicate that the impact absorption energy shows enhanced toughness compared to the research by Balasubramanian et al. [36]. Furthermore, the study by Cui et al. [41] reveals that although the impact toughness of the weld metal does not surpass that of the base material, the presence of high-angle grain boundaries positively contributes to the impact toughness. This suggests that our welding process may offer optimized conditions that preserve or enhance joint performance without compromising material toughness. (Page 23, Line20- Page 24, Line8)
  • The reference added below:

[39] doi.org/10.1108/GS-11-2019-0052;

[40] doi.org/10.3390/ma16041509; 

[36] doi.org/10.1016/S1003-6326(11)60850-9; 

[41] doi.org/10.1016/j.jmapro.2019.01.031.

  1. The reviewer’s comment: Fig. 11. shows pores, the authors need to do an image analysis to come to this conclusion.
  • Response:We greatly appreciate your insightful suggestion. However, this represents the type of pores found at the bottom of the weld metal, which are pronounced, possibly due to the unclear image. We have optimized the image for better clarity, as listed below. Taking your suggestion into consideration, we have checked and amended the sentences.
  •  
  • 1112 (a3) one of sampled WM as the center
  • In the WM, both the radiative zone and the shear lips feature pores of notably differing sizes (approximately 30-50μm), exhibiting a clustered distribution, particularly as observed in Figure 12(a2-a3). This phenomenon primarily stems from the collapse and closure of keyhole pores during the welding process, culminating in process-induced porosity [37-38]. Such pores are likely to diminish the impact absorption energy of the WM, consequently reducing its toughness and resulting in a divergence from the anticipated optimal weld toughness.(Page 21, Line1-8)
  • Thereference added:

[37] doi.org/10.1016/j.optlastec.2024.110569;

[38] doi.org/10.1016/j.actamat.2012.02.035.

  1. The reviewer’s comment:Fig. 10. Why is hardness higher at the fusion zone? Can the author relate this to microstructure?
  • Response:Thanks for your valuable The increased hardness in the fusion zone (WM here in text) of titanium alloy welding is primarily due to the microstructure formed by rapid cooling during the welding process. In the fusion zone, rapid cooling can lead to the formation of fine, hardened α’martensitic phases. This fine martensitic structure is much harder than the original structure of the titanium alloy, thus increasing the overall hardness of the welded area. Additionally, the welding process may cause solute elements to enrich in the fusion zone, further increasing the hardness. Taking your suggestion into consideration, we have checked and amended the sentences, which are also listed below.
  • The increased hardness in WM of titanium alloy welding is primarily due to the microstructure formed by rapid cooling during the welding process. In the WM, rapid cooling can lead to the formation of fine, hardened α’martensitic phases. This fine martensitic structure is much harder than the original structure of the titanium alloy, thus increasing the overall hardness of the welded area[30]. This is also related to the observed distribution of Widmanstatten structures [31-32]. (Page 16, Line17- Page 17, Line1)
  • Thereference added below:
  • [30]org/10.1016/j.msea.2010.10.070

[31] doi.org/10.1016/j.matchar.2021.111162

[32] doi.org/10.1016/j.jmst.2018.03.012

Reviewer 4 Report

Comments and Suggestions for Authors

This paper describes an experimental study of multi-layer multi-pass welding using a laser-metal inert gas hybrid approach on a Ti alloy. The work is well done with excellent characterization plus the development of a nice model for what is happening. I recommend some minor chanes.

Please try to improve the contrast in Figures 6a, 7a, 7b, all of Figure 8, 13b.

Also make the figures of eh hardness distribution in Figure 11 larger and improve the contrast.

Can they add error bars to Figure 10?

Are the values in column 4 in Table 4 really good to 5 place accuracy? They need to give error bras for these values as well. Also, no decimal place in the averages in column 5 as the range is larger than that.

Overall, a nice study on a novel welding approach with a good model for what is happening.

Can the authors say anything about the maximum and minimum thickness of the Ti-based material that can be welded with their approach?

Author Response

Detailed responses to the reviewers’ comments

Dear editor and reviewers:

Thanks for your positive suggestions on our ‘coatings-2915898’ entitled Microstructure and impact toughness of laser-arc hybrid welded joint of medium-thick TC4 titanium alloy”. We express our appreciation for reviewers’ critical comments and kind suggestions to improve this manuscript. We revised the manuscript in accordance with the reviewers’ comments, and carefully proof-read the manuscript to minimize typographical and grammatical errors. The point-by-point responses to the concerns have been attached below this page, which is given in normal font and changes/additions to the manuscript are given in yellow text with red font. Besides, all the changes are presented in the revised manuscript. 

We believe that the quality of this manuscript has been significantly improved by modifying it based on the reviewer comments. We sincerely hope that all the corrections will meet your requirements and expectations, and look forward to your approval on this publication.

Thank you for your time and consideration.

Yours Sincerely

Peng Luo 

Responses to Reviewer #4:

This paper describes an experimental study of multi-layer multi-pass welding using a laser-metal inert gas hybrid approach on a Ti alloy. The work is well done with excellent characterization plus the development of a nice model for what is happening. I recommend some minor changes.

  1. The reviewer’s comment: Please try to improve the contrast in Figures 6a, 7a, 7b, all of Figure 8, 13b.
  • Response:Thanks for your valuable Taking your suggestion into consideration, we have checked and improved the figures, which are also listed below.

  • 67 (a)The microstructure distribution of the welded joint;

  • 78 Microstructure of WM of the laser-MIG hybrid welded TC4 joint: (a)OM; (b)SEM.

  • 89 Microstructure of the different zone of laser-MIG hybrid welded TC4 joint.(a)WM in 3rd# weld; (b) HAZ in 3rd# weld; (c) WM in 2nd# weld; (d) HAZ in 2nd# weld; (e) WM in 1st# weld; (f) HAZ in 1st# weld.

  • 1314(a)TEM image of the microstructure near the fracture; (b) the diffraction spots

  1. The reviewer’s comment: Also make the figures of eh hardness distribution in Figure 11 larger and improve the contrast.
  • Response:Thanks for your valuable suggestion. Taking your suggestion into consideration, we have checked and improved the figure, which are also listed below.

  • 1112 Macroscopic fracture morphologyof laser-MIG hybrid welded TC4 joint

  1. The reviewer’s comment:Can they add error barsto Figure 10?
  • Response:We appreciate your valuable feedback. In response to your comments, we have performed additional testing and independently generated new illustrations. The revised results are now presented in the subsequent figure. Following your recommendation, we have also reviewed and corrected the sentences, as delineated below.
  •  
  • 11Hardness distribution of laser-MIG hybrid welded TC4 alloy joint:(a)3rd# layer, (b)2nd# layer, (c)1st# layer.

  1. The reviewer’s comment: Are the values in column 4 in Table 4 really good to 5 place accuracy? They need to give error bras for these values as well. Also, no decimal place in the averages in column 5 as the range is larger than that.
  • Response:Thank you for your valuable feedback. The values in column 4 of Table 4 are read out directly by the device, and we have made improvements to consider the problems you raised. In response, we have adjusted the precision of the values in column 4 of Table 4. Additionally, we have revised the averages in column 5 to include appropriate decimal places, aligning with the larger range. Your suggestions have greatly improved the clarity and precision of our data presentation.
  • column 4-5 of Table 4.

Bending force (kN)

Average force (kN)

Standard Deviation

30.05

30.02

0.08

29.90

30.10

31.41

30.96

0.69

31.49

29.98

  1. The reviewer’s comment: Overall, a nice study on a novel welding approach with a good model for what is happening.Can the authors say anything about the maximum and minimum thickness of the Ti-based material that can be welded with their approach?
  • Response:Thanks for your valuable suggestion. In response to your inquiry, we appreciate the positive feedback on our study and your specific question. Within the scope of our current research, we primarily focused on the development and validation of a novel welding approach and its modeling impact on the weldability of Ti-based materials. As such, detailed exploration regarding the maximum and minimum thicknesses of the Ti-based material that can be welded with our approach was not covered in this work. However, we acknowledge the importance of this parameter, particularly critical for welding applications of Ti-based materials. Hence, we consider this as one of the significant directions for future research. We plan to explore in detail the effects of different thicknesses on the weldability in our subsequent works and further optimize our welding approach to accommodate a broader range of material thicknesses. Thank you again for your valuable suggestion, and we look forward to addressing this topic in our future research.

Round 2

Reviewer 1 Report

Comments and Suggestions for Authors

Thanks the authors for their satisfactory revisions.

Reviewer 2 Report

Comments and Suggestions for Authors

The corrections requested and the answers given in the 1st revision of the study titled "Microstructure and impact toughness of laser-arc hybrid welded joint of medium-thick TC4 titanium alloy" increased the scientific and originality value of the article. The article is acceptable in its current state.

Reviewer 3 Report

Comments and Suggestions for Authors

Paper can be accepted.